

# 1 Groundwater impacts on surface water quality and nutrient loads in lowland polder catchments: monitoring the greater Amsterdam area

Liang Yu[1,2], Joachim Rozemeijer[3], Boris M. van Breukelen[4], Maarten Ouboter[2], Corné van der Vlugt[2],
Hans Peter Broers[5]
[1]Life and Earth Sciences, Vrije University Amsterdam, Amsterdam, 1181HV, The Netherlands
[2] Waternet Water Authority, Amsterdam, 1096 AC, The Netherlands
[3]Deltares, Utrecht, 3508 TC, The Netherlands
[4]Delft University of Technology, Delft, 2628 CN, The Netherlands
[5]TNO Geological Survey of The Netherlands, Utrecht, 3584 CB, The Netherlands
*Correspondence to*: Hans Peter Broers (hans-peter.broers@tno.nl)
**Abstract.** The Amsterdam area, a highly manipulated delta area formed by polders and reclaimed lakes, struggles with high
nutrient levels in its surface water system. The polders receive spatially and temporally variable amounts of water and
nutrients via surface runoff, groundwater seepage, sewer leakage and via water inlet from upstream polders. Diffuse
anthropogenic sources, such as fertilizer use and atmospheric deposition, add to the water quality problems in the polders.
The major nutrient sources and pathways have not yet been clarified due to the complex hydrological system in such lowland
catchments combined with both urban and agriculture areas. In this study, the spatial variability of the groundwater seepage
impact was identified by exploiting the dense groundwater and surface water monitoring networks in Amsterdam and its
surrounding polders. Twenty-three variables (concentrations of Total-N, Total-P, $NH_4$, $NO_3$, $HCO_3$, $SO_4$, Ca, and Cl in
surface water and groundwater, seepage rate, elevation, land-use, and soil type) for 144 polders were analysed statistically
and interpreted in relation to sources, transport mechanisms and pathways. The results imply that groundwater is a large
source of nutrients in these mixed urban/agricultural catchments, given the higher nutrient levels in groundwater compared
with surface water. The groundwater nutrient concentrations exceeded the surface water Environmental Quality Standards
(EQSs) in 93 % of the polders for TP and in 91 % for TN. Groundwater outflow into the polders thus adds to nutrient levels
in the surface water. High correlations ($R^2$ up to 0.88) between solutes in groundwater and surface water, together with the
close similarities in their spatial patterns, confirmed the large impact of groundwater on surface water chemistry, especially
in the polders that have high seepage rates. Our analysis indicates that the elevated nutrient and bicarbonate concentrations in
the groundwater seepage originate from the decomposition of organic matter in subsurface sediments coupled to sulfate
reduction and possibly methanogenesis. The large loads of nutrient rich groundwater seepage into the deepest polders
indirectly affect surface water quality in the surrounding area, because excess water from the deep polders is pumped out and
used to supply water to the surrounding infiltrating polders in dry periods. The study shows the importance of the connection
between groundwater and surface water nutrient chemistry in the greater Amsterdam area. We expect that taking account of





groundwater-surface water interaction is also important in other subsiding and urbanising deltas around the world, where
water is managed intensively in order to enable agricultural productivity and achieve water sustainable cities.
**1 Introduction**
The hydrology of many lowland delta areas is highly manipulated by human activities such as ditching, draining, and
embanking, to enable agriculture and habitation. The Netherlands is a densely populated country where surface water
salinization and eutrophication are common problems. It is a typical highly urbanized country, with 2/3 of its land lying
below mean sea level. In The Netherlands, small regulated catchments called polders have been developed over centuries by
diking in and draining lakes and swamps (Huisman, 1998). Over 10 million people are living in the coastal area, mainly in
the Western part where a Holocene layer of peat and clay covers Pleistocene fluvioglacial sands. Especially the deepest
polders receive large amounts of groundwater seepage. The surface water levels within the polder catchments are artificially
controlled by pumping water out into the regional water systems (called Boezem), which further accelerates groundwater
seepage. Some of the deep polders exhibit upconing of deep saline groundwater into the surface water. The salt loading
towards these polders is expected to increase, mainly due to the further lowering of surface water levels in response to
subsidence (e.g. Oude Essink et al., 2010; Delsman et al., 2014). Draining the peat polders has also led to subsidence and
repetitive lowering of surface water and groundwater levels. As a consequence, nutrients are released due to peat oxidation
(Hellmann and Vermaat, 2012). Another nutrient source is the large scale agricultural application of manure and fertilizer.
Although manure legislation was already enforced in 1986, surface water quality in the area still does not meet the EU Water
Framework Directive standards for chemical and ecological water quality (Rozemeijer et al., 2014). The local water
authority, called Waternet, is commissioned to improve water quality in a cost-effective mitigation program. The assessment
of load contributions from different pollution sources is essential to set realistic region-specific water quality targets and to
select appropriate mitigation options.
Influences of groundwater on surface water quality have recently gained more attention by hydrologists (e.g. Rozemeijer and
Broers, 2007; De Louw et al., 2010; Garrett et al., 2012; Delsman et al., 2015). Rozemeijer et al. (2010) found that
groundwater seepage has large impacts on surface water quality in a lowland agricultural catchment. A study by Holman et
al. (2008) in the United Kingdom and the Republic of Ireland also suggested that the groundwater contribution to surface
water nutrient concentrations is more important than previously thought. Furthermore, Meinikmann et al. (2015) found that
lacustrine groundwater discharge contributed for more than 50% of the overall external P load in their study lake.
Vermonden et al. (2009) concluded that upward seepage from Meuse-Waal canal delivered $NO_3$ and Cl to urban surface
water system. The impact of other landscape characteristics on surface water quality, such as soil type and land use, has also
been explored. For example, Van Beek et al. (2007) found that nutrient rich peat layers will remain a potential source of
nutrients in surface water in many peat polders in the western part of The Netherlands. Mourad et al. (2009) found that the
spatial patterns of nitrate and phosphate concentrations in the Ahja River catchment in Estonia were related to spatial



differences in urban and agricultural land use proportions. Vermaat et al. (2010) studied 13 peat polders in the Netherlands and reported that agricultural land use largely determined the variability in nutrient concentrations and loads. Phosphorus was observed in higher concentrations in urban areas than in rural areas by Meinikmann et al. (2015) In some studies, point sources like effluent from sewage treatment plants dominated the phosphorus loads (e.g. Wade et al., 2012), but the Netherlands is known to have early invested in centralised sewage treatment works, thus avoiding the many individual spills that are present is some bordering countries (EU, 2017).

Previous water quality research in polder areas have mainly focused on the impact of land use types and topography. The impact of groundwater and flow routes on spatial water quality patterns in polders has not been systematically studied. Such insight is highly needed, as a cost-effective protection and regulation of water resources requires an integrated assessment of water and contaminant flow routes in the water system as a whole. In general however, water and contaminant flow routes in urban settings are more complex than in rural areas, due to the highly variable surface permeability and human emissions of pollutants.

This study aimed at identifying the impact of groundwater on surface water quality in the polder catchments of the greater Amsterdam city area, which is the management area of Waternet, the organisation which manages dikes, regulates water levels and pumping regimes and is responsible for the clean surface water, drinking water supply and waste water treatment. To achieve this, we analysed regional surface water and groundwater quality monitoring data in combination with eight landscape characteristic variables for 144 polders: surface elevation, paved area percentage, surface water percentage, seepage rate, and soil type represented by calcite, humus, and clay percentages. Our statistical analyses yielded insight into the impact of groundwater on the surface water chemistry of the urban and rural polders of Amsterdam. The presented approach contributes to realistic and effective water quality regulation in the Waternet management area and can also be applied to other deltas in the world with adequate groundwater and surface water monitoring data.

## 2 Methods

### 2.1 Study area

This study focuses on the polder catchment landscape around the city of Amsterdam in The Netherlands. The whole study area spans 700 km$^2$, from downtown Amsterdam situated in the northwest to the border of the province of Utrecht in the southeast (Fig. 1). Amsterdam is a low-lying highly paved city located in the western part of the Netherlands, developed around the levees of the tidal outlet of the Amstel River about 700 years ago (Vos, 2015). Nowadays, the water system in Amsterdam is connected to the large fresh water body of the Lake IJ (Fig. 1). Besides Lake IJ, other important large water bodies are the Amstel and Vecht rivers and the Amsterdam-Rhine canal. This regional water system, also called the 'Boezem', connects the Amsterdam-area water system to the Rhine River (upstream) and the Lake IJssel and the North Sea (downstream). In the 19th and 20th centuries, the city expanded, and many new neighbourhoods and suburbs were built.



Polders and reclaimed lakes form the main landscape in the southward extensions of the city. Some of these polders are at
several meters below Mean Sea Level (MSL) and are influenced by groundwater seepage.
**2.1.1 Landscape history and hydrology**
**Landscape history**
Our study area is located in the western part of The Netherlands where large rivers and the sea have intensively interacted for
millions of years. The main topographic feature is a Pleistocene sandy ice pushed ridge with elevations ranging from 0 to 30
m, which is located on the east part of the study area (Fig. 1, Fig. S1). To the west, the ridge is bordered by the broad
periglacial Pleistocene river plains of the Rhine delta. During the Holocene, these sandy river plains were covered with peat
and clay, which are currently found at the surface throughout the western part of the Netherlands, on top of Pleistocene
sands. The average thickness of the Holocene peat and clay cover is 20 m, although it increases to over 50 m in former tidal
inlet channels (Hijma, 2009).
In 1000 AD, about 5000 years after first settlers appeared in these low lands, the inhabitants started mining peat, digging
ditches, constructing dikes, reclaiming former swamps and lakes, and pumping water out into a large scale drainage system
(called Boezem). Special hydrological catchments called 'polders' were formed, connected by the 'Boezem' main waterways
around them. Fig. S1 and and Table S1 in the Supplementary Information show the entire system of polder catchments
(indicated by numbers for reference) and boezems studied in this paper. Prominent on these maps are two deep polders
Horstermeer (#79) and Groot Mijdrecht (# 80), two former lakes that were formed after peat excavations. Drainage for lake
reclamation and groundwater extraction (Schot, 1992a) caused further subsidence and increased seepage of paleo-marine
brackish groundwater from deep aquifers (Delsman et al., 2014).
The long history of marine influence stopped after closing off the estuaries and the inland sea in the 20[th] century ( Huisman,
1998). In 1932, the construction of the Closure Dike (Afsluitdijk) created the fresh water Lake IJssel out of the former salt
water Zuiderzee ('Southern Sea') to protect the surroundings from floods and to enable land reclamation. The former marine
impact is still reflected by the presence of brackish groundwater in the shallow subsurface (Schot, 1992 b).
The construction of the Amsterdam Rhine Canal separated the study area into two parts (Fig. S1): the Central Holland in the
west and the Vecht lakes area in the east. In the Central Holland polders, relatively thick peat layers and pyrite rich clays are
still present in the shallow subsoil, as described by Van Wallenburg (1975). The Vecht lakes area is characterized by large
open water areas and a number of wetland nature reserves. The rest of the Vecht lakes area is mainly grassland used for dairy
farming. Soils in this area are generally wet and rich in organic matter and clay (Schot, 1992 b).
Mainly during the 20[th] century, the urban areas have been growing from the historic city-centres on river and tidal channel
levees into the surrounding low-lying polders. To facilitate the construction of buildings, a 1-5 meter-layer of sand was often
supplied on top of the original sediments. The thickness of this suppletion sand layer is extremely variable even at a small
scale. The sand suppletions are either calcite-poor without shell fragments or calcite rich with shell fragments that indicate





their (peri-) marine origin. The spatial distribution and sources of the sand suppletions probably influence groundwater and
surface water chemistry, but are poorly registered.
**Polder hydrology**
Within the polders, the water levels are artificially maintained between fixed boundary levels to optimize conditions for their
urban or agricultural land use. Boezem water levels always exceed the polder surface water levels.  In the case of water
deficiency, water is let into the polder ditches from the boezem through pipes by gravity flow (Fig. 2a). Pumping stations are
situated at the boezems to regulate the water levels in the polders in times of precipitation excess. In the case of a water
surplus, pumping stations start pumping water out of the polder into the boezem-system (Fig. 2b).
The regional flow directions in wet and dry periods in the study area are depicted in Fig. 3. The Amsterdam-Rhine canal, the
Amstel River, and the Vecht River are the main water courses discharging surface water from the south to the north in
periods of water surplus (Fig. 3). In periods of water deficiency, however, the flow directions are reversed in some parts of
the system.
There are six main sources of inlet water to compensate for water shortage in dry periods (Fig. 3): (1) Amsterdam Rhine
Canal (ARC): water of the ARC originates from the Rhine and is supplied as inlet water for the southeast polders and
polders in the southeast of Amsterdam city; (2) Amstel River: the historic canals of the city of Amsterdam are mainly
flushed by water from the Amstel River. Via the canals, this water discharges to the downstream part of the ARC and further
into the North Sea; (3) Groot Mijdrecht and Horstermeer: the brackish surplus of seepage water from the deep polder Groot
Mijdrecht (~1000 mg Cl $L^{-1}$ on average) and Horstermeer (~500 mg Cl $L^{-1}$) is pumped into the Boezem system and is
redistributed towards surrounding polders (pink lines in Fig. 3); (4) Rijnland Water Authority district: polders in the far west
of the study area receive inlet from the neighboring Water Authority district Rijnland. The water quality of this source is
unknown. (5) and (6) Vecht River and Lake IJ: polders along the Vecht River receive inlet water that partly originates from
the Rhine and partly from Lake IJ. Polders close to Lake IJ receive large amounts of water directly from the lake. The Lake
IJ water is also used to flush canals in the city of Amsterdam.
**2.1.2 Characterisation of regions**
Based on the geology and paleohydrological history as introduced in section 2.1.1, 5 regions were identified (see Fig. 4). The
5 regions are: (1) the Zuiderzee margin region, with shallow brackish groundwater, lies directly adjacent to the former salt
water Zuiderzee, which was dammed in the 1930s and transformed into the fresh water Lake IJssel (connected to Lake IJ),
which is now the biggest fresh water reservoir of The Netherlands; (2) the deep polders Groot-Mijdrecht and Horstermeer,
which are reclaimed lakes with clayey lake sediments at the surface. These polders are characterized by upconing of salt
groundwater from deeper layers (Oude Essink et al., 2005; Delsman et al., 2014); (3) the Central Holland system of boezems
and polders which are characterized by a relatively thick sequence of marine clays and intercalated peats; (4) the Vecht lakes
region at the western margin of the ice pushed ridge, characterized by shallow peat soils over a sandy subsoil and large
shallow lakes and wetlands resulting from peat excavations (van Loon, 2010); and (5) the Ice pushed ridge in the eastern part





of the study area, which is characterized by permeable sandy soils, recharge of freshly infiltrated water, and the mere absence of draining water courses.

Our a priori expectation was that the groundwater quality of these 5 regions is significantly different, because of their specific paleohydrological situations and present day groundwater flow patterns. We therefore used the regions to evaluate the groundwater quality patterns and to give structure to our comparisons between groundwater and surface water concentrations and loads.

## 2.2 Data processing

The database that was compiled and used for this study covers 144 individual polders and includes monthly surface water quality data, spatiotemporally averaged groundwater quality data (TN, $NO_3$, $NH_4$, $SO_4$, TP, Ca, $HCO_3$, and Cl), daily pumping station discharge time series, and 15 polder averages of the following statistic variables: polder seepage rates, elevations, surface water and paved area percentages, and calcite, clay, and humus percentages of the upper soil layer. More information about the data processing and the database can be found in the Supplementary information.

### 2.2.1 Groundwater data

A total of 802 observation wells of groundwater quality are available from the period 1910-2013, largely drawn from the National Groundwater Database DINO (TNO, DINOLoket). We selected analyses from the upper 50 m of the subsurface, which corresponds with the thickness of the first main Pleistocene aquifer in the area and the Holocene cover layer. For our analyses, we averaged concentrations at individual monitoring screens of each monitoring well for all sampling dates available.

To analyse the spatial pattern of groundwater quality, we averaged concentrations of all the monitoring wells located in the same polder (for more details, see Table S2). For 24 polders out of the polders without groundwater quality data, the concentrations were estimated by Inverse Distance Weighted interpolation, however using absolute elevation difference instead of distance. The greater the absolute elevation difference, the less influence the polder has on the output value. The equations are:

$$C_0 = \sum_{i=1}^{n} \lambda_i C_i \tag{1}$$

$$\lambda_i = d_{i_0}^{-p} \Big/ \sum_{i=1}^{n} d_{i_0}^{-p} \,, \qquad \sum_{i=1}^{n} \lambda_i = 1 \tag{2}$$

$C_0$, prediction of target polder; $C_i$, observed value of surrounding polders; n number of observations; p, power parameter (2 in this case); $d_{i_0}$, absolute elevation differences of target polder with surrounding polders. Subsequently, to interpret the





groundwater quality patterns, the variation of concentrations in and between the 5 regions was visualized using boxplots
(Helsel and Hirsch, 2002).
Because our dataset contains both fresh and brackish to saline water, we used the mass $SO_4/Cl$ ratio of the samples as an
indicator of sulfate reduction. $SO_4/Cl$ ratios lower than the sea water ratio of 0.14 (Morris and Riley, 1966) point to the
occurrence of sulfate reduction (Appelo and Postma, 2005; Griffioen et al., 2013). Ratios above 0.14 point to the addition of
sulfate relative to diluted sea water through processes like pyrite ($FeS_2$) oxidation or through input via atmospheric inputs,
fertilizers, manure, or leakage and overflow of sewer systems.
Average concentrations in groundwater for each polder were mapped to be compared with average annual surface water
concentrations (See section 2.2.2). The potential relationship between the solute concentrations in groundwater (TN, $NO_3$,
$NH_4$, $SO_4$, TP, Ca, $HCO_3$, and Cl) and the landscape variables (paved area percentage, elevation, seepage rate, surface water
area percentage, lutum, humus and calcite percentages of top soil) were explored using the Spearman correlation, which
reduces the influence of outliers and yields a robust correlation statistic (Helsel and Hirsch, 2002).
To further explore the statistical relations in our data set, scatter plots were made to evaluate $HCO_3$, $SO_4$, Cl, and nutrient
($NO_3$, $NH_4$ and $PO_4$) concentrations in groundwater. We also explored the links between alkalinity (over 99 % of our
groundwater alkalinity was dominated by $HCO_3$, (Stuyfzand, 2006)), Cl concentration, $SO_4/Cl$ ratio and nutrients (TN, $NH_4$
and TP) concentrations. For our interpretation, we also used the calculated amount of consumed or produced $SO_4$ in mg $L^{-1}$
relative to the $SO_4/Cl$ ratio of diluted seawater, using Eq. (3):
$$SO_4\,consumed(-)\;or\;produced(+) = SO_4\,measured - Cl\;measured \cdot SO_4\,sea/Cl\,sea \qquad (3)$$
In order to understand the impact of cation exchange processes involving Ca and Na exchange during salinization and/or
freshening of aquifers (Griffioen, 2004; Stuyfzand, 2006) we defined the amount of exchange $Na_{ex}$ as:
$$Na_{ex} = Na_{gw} - Cl_{gw}(Na_{gw}/Cl_{seaw}) \qquad (4)$$
Where, $Na_{ex}$ is the amount of Na exchange; gw, ground water; seaw, seawater. $Na_{ex} > 1$ points to freshening, $Na_{ex} < -1$ to
salinizing conditions.
**2.2.2 Surface water data**
Loads represent the contribution of polders to surface water quality of the regional water system in weight per time unit. To
eliminate the impact of the size of polders, we calculated daily load per area in kg $ha^{-1}$ $d^{-1}$. This was calculated using the
daily average loads of each solute divided by the polder areas using Eq. (5):

$$\text{Load per area} = \frac{L}{A} = \frac{1}{A} \cdot \frac{C \cdot Q}{1000} \qquad (5)$$

Where L is daily load kg $d^{-1}$, A is polder area (ha), C is daily solute concentration in mg $L^{-1}$ and Q is daily discharge in $m^3$ $d^{-1}$
$^{1}$. Average daily loads for each year were multiplied by 365 to get average yearly loads per area. Monthly surface water





quality measurements for the period 2006-2013 of 144 polders were extracted from the Waternet database. The
measurements were converted to daily time series by stepwise interpolation between the monthly measurements. We
assigned a concentration of zero to measurements below the detection limits. Discharge data Q are daily measurements over
the same time period. An average over multiple pumps, when present, was taken for each polder. For further details about
the data processing we refer to Table S2.
The pumping discharge is regulated to respond to water surplus or deficiency conditions in the polder catchments. Using the
pumping frequency data, we proved that solute concentrations in pumped water are usually higher at the beginning of each
pumping activity (Van der Grift et al., 2016). The pumping rates may also influence water quality in the polder. To eliminate
differences caused by pumping rates, we used the normalized concentration calculated using Eq. (6):

$$C = \frac{\text{Load per area} \cdot A}{Q} \qquad (6)$$

In this equation, C is the normalized concentration (mg $L^{-1}$), Load per area is from Eq. (1), Q is the pumping discharge ($m^3 \, y^{-1}$
), and A is the polder area ($m^2$).The statistical methods that were used for groundwater quality (described in section 2.2.1)
were also applied to the surface water normalized concentrations.
Based on a national assessment on ecosystem vulnerability, Environmental Quality Standards (EQSs) were set by the Water
Boards (Heinis and Evers, 2007). For most ditches and channels in the clay and peat regions, EQSs of TN and TP are 2.4 mg
$L^{-1}$ and 0.15 mg $L^{-1}$, respectively (Rozemeijer, 2014). We used these most common EQSs as reference concentration values.
For example, the EQSs of TN and TP were used for the legend classifications in our surface water quality maps and were
added as reference lines in our concentration boxplots. Percentages of polders exceeded these standards were calculated in
this paper.
**2.2.3 Surface water compared with groundwater solute concentrations**
We statistically analysed the groundwater and surface water quality data and landscape characteristic variables by (1)
calculating the correlation coefficients between averaged groundwater solutes concentrations and normalized concentrations
of surface water using the Spearman method, and (2) by selecting variables (based on the correlation matrix above) to be
integrated into multiple linear regression models for predicting surface water solute concentrations. Again, the Spearman
method was applied and linear regression was based on ranks in order to avoid outliers to determine the outcomes. The
explaining variables for surface water concentrations include groundwater solute concentrations, landscape characteristics,
and the $SO_4/Cl$ ratio in groundwater. The number of selected explaining variables depends on the added value of an extra
component. An extra explaining variable was only added to the regression when it improved the explained variance with at
least 5%, and at most four variables were added. For comparison purposes, we also used the surface water EQSs as reference
concentration values in the groundwater quality maps and boxplots, although the EQS's have no administrative meaning for
groundwater itself.



**2.2.4 Solutes redistribution in surface water**
Loads were used to assess the impact of different polders as sources of solutes for the boezems and the receiving water
bodies further downstream. In general, the spatial patterns can be distinguished through maps of the surface water solute
loads per area if there are no other influences. However, there are exceptions such as the seepage water which is pumped out
of the two upconing polders Groot Mijdrecht and Horstermeer, which is discharged into the Boezem system and used as inlet
water for the surrounding polders during summer. To show the impact of this inlet water on the receiving polders' water
quality, we analysed the inlet solute loads and the resulting surface water concentrations for polder Botshol. Polder Botshol
(part of polder # 104 Noorderpolder of Botshol (zuid and west)) with an area of 1.3 $km^2$ receives inlet water from the Amstel
boezem system that has a significant contribution of seepage water that is pumped out of the polder Groot Mijdrecht.
Two models were applied for simple solute concentration calculations based on inlet water quality. Model 1 calculates the
accumulation of solutes in the water body, with evaporation as the only output for water (leaving the solutes behind). Model
2 models the complete mixing and outlet of both water and solutes via other routes like the outlet weir, infiltration, and
leakage. In reality, water leaves Polder Botshol partly via evaporation (Model 1) and partly via other routes (Model 2):
Model 1 (evaporation): $\quad\quad C_{i+1} = (C_i \cdot V_0 + C_{inlet} \cdot Q_{inlet})/V_0$ $\quad\quad\quad\quad\quad\quad\quad\quad\quad$ (7)
Model 2 (infiltration/outlet): $\quad C_{i+1} = (C_i \cdot V_0 + C_{inlet} \cdot Q_{inlet})/(V_0 + Q_{inlet})$ $\quad\quad\quad\quad$ (8)
where, $C_{i+1}$ is the predicted solute concentration after getting inlet water at time i; $C_i$ is the predicted solute concentration in
the polder at time i, the outlet measurements in the beginning of wet period were taken as $C_0$; $V_0$ is the water volume in the
polder (800.000 $m^3$), which is assumed to be constant as water levels are tightly controlled; $C_{inlet}$ is the estimated Cl
concentration in the inlet water (1000 mg $L^{-1}$); $Q_{inlet}$ is estimated constant inlet water volume, 6000 $m^3$ $d^{-1}$. All parameters
are shown in supplementary information Excel spread sheets. The models were applied in the year 2006, 2008, 2009, 2010,
2011 and 2012.
**3 Results**
**3.1 Spatial pattern and statistical analysis of groundwater quality**
Fig. 5 and Fig. 6 show the groundwater quality for the upper main aquifer under the 144 polders for Cl, Ca, $HCO_3$, $SO_4$, TN,
$NH_4$, $NO_3$, and TP. The relations between groundwater solutes, landscape variables, and potential hydrochemical reactions in
the subsurface were explored by correlation analysis of which the results are shown in Table 1 (yellow part) and Fig. 7 and
Fig. 8.
**Cl, Ca, and $HCO_3$**
In Fig. 5, the Zuiderzee margin, where brackish groundwater is dominant, P25 and P75 of concentrations are between 290
and 2100 mg $L^{-1}$ Cl, 100-300 mg $L^{-1}$ Ca, and 400-1000 mg $L^{-1}$ $HCO_3$. Relatively high concentrations of Cl, Ca and $HCO_3$





were also for the two deep polders Groot Mijdrecht (# 80) and Horstermeer (Upconing area) with known upconing of salt
groundwater. The Central Holland area was dominated by fresh groundwater with low Cl and Ca concentrations, but with
considerable amounts of $HCO_3$. Polders with relatively high chloride (>1000 mg $L^{-1}$) are distributed along the former
Zuiderzee margin, plus the Upconing area which is two deep polders with known upconing of brackish water. Relative to the
regions above, the Vecht lakes area and the Ice pushed ridge showed significantly less mineralized waters with lower $HCO_3$
and Cl concentrations. For example, the P75s of Cl in these two regions are below 150 mg $L^{-1}$ and the P75s of $HCO_3$ below
350 mg $L^{-1}$. The groundwater $HCO_3$ concentrations (Fig. 6) show an east-west increasing trend with highest concentrations
in both the fresh and brackish areas west of the Amsterdam Rhine Canal.
**$SO_4$ and $SO_4$/Cl**
The Zuiderzee margin and the Upconing area showed large ranges of $SO_4$ concentrations (P25 and P75: 7-125 mg $L^{-1}$ and 7-
250 mg $L^{-1}$, respectively) with the $SO_4$/Cl mass ratios generally lower than the 0.14 ratio for diluted seawater. The polders in
the eastern Zuiderzee margin showed the highest average $SO_4$ levels (Fig. 6). The Central Holland area exhibited the lowest
$SO_4$ concentrations with the smallest variability, with $SO_4$/Cl P75 typically lower than 0.14. However, some outliers in this
region reached quite high sulfate concentration levels (>200 mg $L^{-1}$). The Vecht lakes and the Ice pushed ridge showed
intermediate sulfate concentrations and typically have a $SO_4$/Cl ratio clearly above 0.14.
**$NH_4$, TN, $NO_3$, and TP**
The higher groundwater $NH_4$ and TP concentrations generally locate in the western part of the study area (Zuiderzee margin,
Upconing area, and Central Holland regions). Median $NH_4$ concentrations in the Zuiderzee margin (6.4 mg $L^{-1}$) and Central
Holland (10.6 mg $L^{-1}$) were far higher than in the Vecht lakes (2.1 mg $L^{-1}$) and Ice pushed ridge regions (0.07 mg $L^{-1}$). The
same was observed for TP (0.7, 1.6, 0.2 and 0.06 mg P $L^{-1}$, respectively). Nutrient concentrations in the Upconing area
(medians 5.7 mg $NH_4$ $L^{-1}$ and 0.14 mg P $L^{-1}$) were relatively low compared with the groundwater in the Zuiderzee margin
and Central Holland areas, although we consider the $NH_4$ concentration levels to be substantial given the surface water EQS
of 2.4 mg N $L^{-1}$. TN showed the highest median concentration levels in the Zuiderzee margin and Central Holland regions, as
well as in the Ice pushed ridge (7.3 mg N $L^{-1}$). The Ice pushed ridge region also showed the highest level of $NO_3$. In the latter
region, nitrate is the main component of TN, while $NH_4$ is the main component in the other regions.
Groundwater quality varied from fresh, low mineralized in the eastern parts (Vecht lakes and Ice pushed ridge, Figure 4)
towards brackish, highly mineralized and nutrient rich groundwater in the northwest (Zuiderzee margin and Central Holland,
Fig. 4). This relationship was further indicated by the strong correlations between Ca and Cl (Spearman $R^2$ 0.77) and
between $HCO_3$, TP and $NH_4$ ($R^2$ 0.68-0.82) (Table 1, yellow part). The spatial Ca pattern corresponds largely with the Cl
pattern (Fig. 6), showing higher Ca concentrations in the brackish waters, which is related to the high Ca concentrations in
(diluted) seawater (Section 4.1). The strong correlation between TN and $NH_4$ ($R^2$ 0.81) showed the dominance of $NH_4$ in TN,
except in the suboxic groundwaters under the Ice pushed ridge where nitrate dominates TN. $HCO_3$, TP and $NH_4$ were all
weakly negatively correlated with elevation, indicating that higher concentrations exist in the deeper polders which are more
affected by brackish groundwater seepage.





In the more mineralized groundwater systems, sulfate reduction is a potential cause of the significant relationship between
$HCO_3$, TP, and $NH_4$. From using the $SO_4/Cl$ ratio of the samples and comparing them with the $SO_4/Cl$ ratio in seawater (Eq.
3), it appears that most of the brackish groundwater showed signs of sulfate reduction. Figure 7 shows that the amount of
$SO_4$ consumed in the sulfate reduction process increased with the chloride concentration of the groundwater, and that sulfate
reduction was complete only in part of the groundwaters. Note that groundwater below polders with excess $SO_4$ are all in
water with Cl<1000 mg $L^{-1}$. It follows from Figure 8 that high $HCO_3$, TP, and $NH_4$ concentrations mostly occurred in
groundwater with a $SO_4/Cl$ ratio lower than 0.14, indicating sulfate reduction which induces the release of N and P from the
mineralized organic matter in the subsurface and the production of alkalinity during that process. Therefore, these waters
typically have increased $HCO_3$ concentrations above 480 mg $L^{-1}$ (Fig. 8A and 8B) and are often associated with brackish
groundwater that once contained sulfate (Fig. 8C: Cl>300 mg $L^{-1}$). The hypothetical chemical relation between sulfate
reduction ($SO_4$ consumed) and $HCO_3/NH_4/H_3PO_4$ production from the mineralisation of organic matter can be found in the
reaction equation below (Stuyfzand, 2006):

$$2SO_4^{2-} + 3.5CH_2O(NH_3)_x(H_3PO_4)_yI_zBr_a + Fe^{2+}$$
$$\rightarrow FeS_2 + (2 + 3.5x)HCO_3^- + (1.5 - 3.5x)CO_2 + 3.5xNH_4^+ + 3.5yH_3PO_4 + zI^- + aBr^- \quad [1]$$

**3.2 Spatial patterns and statistical analysis of surface water quality**
Fig. 9 and Fig. 10 show the solute concentrations in the four regions: Zuiderzee margin, Upconing area, and Central Holland,
and Vecht lakes. Due to insufficient surface water quality data, no results are shown for several polders in the Amsterdam
city area (see Fig. 4) and the Ice pushed ridge region. The first is related to the monitoring priorities of the Waternet water
board, the latter is related to the almost absence of surface water in this region.
**Cl, Ca, and $HCO_3$**
Highest chloride levels (>300 mg $L^{-1}$) were found in the Upconing polders with brackish seepage and in a minority of the
polders in the Zuiderzee margin and Central Holland regions (Fig. 9 and 10). The high Ca and $HCO_3$ concentrations in these
polders are also related to the occurrence of brackish water. However, most of the surface water in the Zuiderzee margin and
the Central Holland area is fresh with relatively low Cl concentrations (Fig. 10). The Vecht lakes area exhibits the most fresh
and least mineralized surface water.
**$SO_4$ and $SO_4/Cl$**
The highest $SO_4$ concentration levels and $SO_4/Cl$ mass ratios mostly occurred in the Central Holland area, especially the
western part. The elevated $SO_4$ and $SO_4/Cl$ ratios indicate the presence of sulfate sources other than (relict) seawater in this
area, probably atmospheric deposition, agriculture and/or oxidation of pyrite exposed in the upper soils which developed in
marine clay deposits and are denoted as "cat clays" (Wallenberg, 1975). In the Zuiderzee margin and the two upconing
polders, the median $SO_4$ levels are 64 and 62 mg $L^{-1}$, respectively, and $SO_4/Cl$ mass ratios of the two upconing polders are
below 0.14. A generally lower $SO_4$ with $SO_4/Cl$ ratios far exceeding the 0.14 were found in the Vecht lakes region.





**TN, NH$_4$, NO$_3$, and TP**
According to Fig. 9 and Fig. 10, surface water EQSs of TN (2.4 mg N L$^{-1}$) and TP (0.15 mg P L$^{-1}$) were exceeded in most
polders of the study area. The outliers with even higher nutrient concentrations are mainly located in the west of the Central
Holland region. P25 and P75 of TP and TN in the Zuiderzee margin and in Central Holland regions all significantly
exceeded EQSs for surface water. In the two upconing polders, polder Groot Mijdrecht showed higher concentrations of TP
and TN than polder Horstermeer (medians of 0.28 vs. 0.11 mg P L$^{-1}$ and 5.4 vs. 1.8 mg N L$^{-1}$). Polders with concentrations
below the EQS's were mainly situated in the Vecht lakes area where large open water areas exist. In this region, TP slightly
exceeded the EQS with a median concentration of 0.22 mg L$^{-1}$, while the median TN concentration of 2.26 mg L$^{-1}$ was just
below the EQS. The concentrations of NO$_3$ and NH$_4$ in the Vecht lakes area were relatively low as well.
Similar to the results of groundwater, higher nutrient levels also exist in higher mineralized surface waters, which is also
indicated by the correlation results (Table 1, blue part): Ca and HCO$_3$ are both correlated with NH$_4$ (Spearman R$^2$ are 0.64
and 0.67), TP (R$^2$ 0.55, 0.62), and TN (R$^2$ 0.57, 0.47). In surface water, Ca and HCO$_3$ had a significant correlation (R$^2$ 0.88).
This indicates that groundwater is the probable source of the water and nutrients in the surface water of the polders. This
groundwater impact was further supported by the correlations between the following pairs of solutes in surface water: Cl
with Ca (R$^2$ 0.55), HCO$_3$ (R$^2$ 0.52), SO$_4$ (R$^2$ 0.49) and NH$_4$ (R$^2$ 0.51), as well as SO$_4$ with TN (R$^2$ 0.57) and NO$_3$ (R$^2$ 0.50). A
more direct indication for the groundwater impact is that NH$_4$, HCO$_3$ and Ca concentrations in surface water were positively
related to the seepage rate. In a similar way, the groundwater impact is suggested by the negative correlations between
elevation and the concentration levels of most surface water solutes (TN: R$^2$ -0.67, NH$_4$: R$^2$ -0.59, NO$_3$: R$^2$ -0.40, HCO$_3$: R$^2$ -
0.48, SO$_4$: R$^2$ -0.47 and Ca: R$^2$ -0.57).
For the soil variables (lutum, humus and calcite), only humus showed correlations with TN, NH$_4$, Ca, and Cl in surface water
(Table 1). Paved area percentage, surface water area percentage, calcite and clay percentages did not show correlation
coefficients above 0.4 with surface water quality. Surface water TN correlated more closely to NH$_4$ (0.77) than to NO$_3$
(0.57), which reflects that NH$_4$ is generally the main form of TN in the study area.
**3.3 Groundwater and surface water quality comparison**
A common spatial pattern in surface and groundwater chemistry is that polders in the Zuiderzee margin area, the two
upconing polders, and the Central Holland area suffer from a worse water quality situation than the polders in the Vecht
lakes and Ice pushed ridge areas. However, compared with the underlying groundwater quality, surface water in the whole
area has much lower chloride, bicarbonate, and nutrient levels, but higher SO$_4$ concentrations (Fig. 5 and Fig. 9). The polders
generally have much higher TP and TN concentrations in groundwater than in surface water. The groundwater nutrient
concentrations exceeded the surface water EQS's in 93 % of the polders for TP, and in 91 % for TN. Polders with
groundwater nutrient levels below the EQS's were mainly found near Lake IJssel. Especially the groundwater TN
concentrations in the Ice pushed ridge severely exceeded surface water EQSs, which can be mainly attributed to the elevated





NO$_3$ concentrations. For TP in groundwater, the Zuiderzee margin and Central Holland areas show more significant EQS
exceedances compared to the Upconing area, Ice pushed ridge and the Vecht lakes area.
Table 1 shows that TP, NH$_4$, HCO$_3$, and Cl concentrations in groundwater correlate with the same components in surface
water (R$^2$ 0.53, 0.43, 0.66, and 0.72). In addition, HCO$_3$ in groundwater showed moderate correlations with nutrient
concentrations in surface water (TP (R$^2$ 0.64), TN (R$^2$ 0.50), and NH$_4$ (R$^2$ 0.46)). HCO$_3$ concentrations in surface water also
correlated with nutrient concentrations in surface water (TP (R$^2$ 0.60), TN (R$^2$ 0.49), and NH$_4$ (R$^2$ 0.59)). Surface water SO$_4$
weakly correlated to groundwater Cl (R$^2$ 0.47).
Based on these correlations, we selected groundwater parameters and landscape characteristics to be integrated in multiple
linear regression models to predict concentrations of surface water components. For most solutes (TP, NH$_4$, TN, HCO$_3$, and
Cl, the R$^2$ of the regression models is around 0.5, which indicates that around 40 ~ 50% of the spatial variance in surface
water can be explained by specific groundwater chemistry parameters, seepage, and elevation. For NO$_3$ and SO$_4$, the R$^2$ of
the regression models (inverse with Elevation) are very low, 0.18 and 0.21, respectively. For all other parameters, the
groundwater HCO$_3$ concentration was the best explaining variable for the surface water concentrations. The spatial variation
in HCO$_3$ $_{SW}$ and Ca $_{SW}$ were relatively well explained by only HCO$_3$ $_{GW}$ combined with Seepage, respectively (Eq. 13 and Eq.

15 15).

The regression models were significantly improved by including groundwater concentrations of TP, NH$_4$, and Cl (Eq. 9, 11
and 16). In regression models Eq. 9, 10, 11, 12, 14, and 15, the elevation of the polders also explained part of the spatial
variation in surface water concentrations. When only including polders with net groundwater seepage, the R$^2$ improved
significantly for TP, NH$_4$ and HCO$_3$.
The results above strongly suggest that the groundwater composition puts limitations to the compliance of the receiving
surface water towards the EQS defined for N and P.

## 3.4 Surface water solute redistribution

Figure 11 showed that the solute loads of polders to the boezem are relatively high in the Zuiderzee margin, the Upconing
polders, and the Central Holland regions. The Vecht lakes area has large open water areas and showed the lowest loads to the
boezem system.
A clear similarity between the spatial patterns of the solute loads and the average seepage rate patterns was observed in Fig.
3 and Fig. 11. In general, polders with high seepage rates also discharge relatively high loads to the Boezem system. Some
examples of polders with relatively high seepage rates are polder #119 (Bethunepolder, 13 mm d$^{-1}$), #79 (Horstermeer, 8.7
mm d$^{-1}$), #50 (Polder De Toekomst, 2.4 mm d$^{-1}$), #131 (Hilversumse Meent, 2.4 mm d$^{-1}$), #98 (Polder Wilnis-Veldzijde, 3.7
mm d$^{-1}$), #80 (Polder Groot Mijdrecht en Polder de Eerste Bedijking (oost), 5.0 mm d$^{-1}$), #74 (Polder de Nieuwe Bullewijk
en Holendrechter- en Bullewijker Polder noord, 1.8 mm d$^{-1}$) and #75 (Bijlmer, 2.0 mm d$^{-1}$). The highest loads are discharged
from the two upconing polders: Groot Mijdrecht (#80) and Horstermeer (#79).



The influence of the redistribution of the large water volumes and loads from deep polders was also observed in Fig. 3 and
Fig. 11. Polders that receive inlet water from Groot Mijdrecht and Horstermeer (see section 2.1.1, Fig. 3) showed relatively
high solute loads, independent of their own seepage or infiltration fluxes. This especially holds for polders downstream of
Groot Mijdrecht and Horstermeer, like polder #73 (Holendrechter- en Bullewijker Polder (zuid en west), -0.05), #74 (Polder
de Nieuwe Bullewijk en Holendrechter- en Bullewijker Polder noord, 1.8 mm d$^{-1}$), #104 (Noorderpolder of Botshol (zuid en
west), -1.4 mm d$^{-1}$), #105 (Noorderpolder of Botshol (Nellestein), -0.7 mm d$^{-1}$), #106 (Polder de Rondehoep, -1.1 mm d$^{-1}$),
and polder #107 (Polder Waardassacker en Holendrecht,  -0.15 mm d$^{-1}$).
The impact of this redistributed water on polder water chemistry is demonstrated by a simple water and solute mass balance
calculation for the receiving polder Botshol (see paragraph 2.2.4). Fig. 12 gives the chloride concentration results of both the
'evaporation' and the 'infiltration/outlet' models. Figure 12 shows that a very simple model can easily explain the peak Cl
concentrations in the Polder Botshol to be the result of the inlet of water from the boezem and Groot Mijdrecht. The
'evaporation' model performs better in 2006 and 2008 and the 'infiltration/outlet' model in 2011 and 2012. Most of the time,
the measured concentrations are between the calculated concentrations from both models. This aligns with the understanding
that water leaves Botshol via a combination of evapotranspiration and other outflow routes, such as infiltration, leakage, and
outlet.
**4 Discussion**
This study aimed at identifying the impact of groundwater on surface water quality in the polder catchments of the greater
Amsterdam city area. According to the statistical analysis of data over five regions in the study area, a clear influence was
identified. Solute concentrations in groundwater and surface water correlated well, although groundwater solute
concentrations were generally much higher than normalized concentrations in surface water. The latter seems logical given
the dilution of surface water by the precipitation surplus on an annual basis, with the annually discharged surface water being
a mixture of seeping groundwater and precipitation. Moreover, similar spatial patterns in solute concentrations were found in
groundwater and surface water. Polders that are influenced by groundwater seepage or by redistributed seepage water from
nearby deep polders are at risk of non-compliance, as groundwater concentrations exceeded the TN and TP EQSs for surface
water in more than 90% of the polders. Consequently, the groundwater nutrients input hinders achieving water quality
targets in the surface water in those lowland landscapes.
**4.1 Key hydro chemical processes**
In general, the groundwater chemistry corresponds with the geological history of the study area. In the peat land polder
catchments within the Dutch delta system of marine, peri-marine and fluvial unconsolidated deposits, abundant organic
matter is present in the subsurface (e.g. Hijma, 2009). The presence of reactive organic matter in the shallow subsurface



depletes the infiltrating groundwater from oxygen and nitrate, leading to an overall low redox potential in groundwater,
which enables the further decomposition of organic matter downstream.
Our data strongly suggests that sulfate reduction, sometimes in combination with methanogenesis, is the main process
releasing nutrients (N, P) and $HCO_3$ from the organic rich subsurface in the study area, especially in both the fresh and
brackish groundwater of the Zuiderzee margin, the Upconing polders, and Central Holland that are charactreized by low
$SO_4/Cl$ ratios (Table 1, Fig. 8). The Holocene marine transgression undoubtedly influenced the chemistry of groundwater by
salinizing processes that also increased sulfate availability derived from diluted sea water. Refreshing of the aquifers by
infiltration of fresh water from rivers and rain in more elevated polders and lakes further influenced part of the groundwater.
We examined the amount of freshening and salinization using the exchange Na ($Na_{ex}$) and investigated how this process may
have influenced the release of P as was suggested by Griffioen et al. (2004). Figure S2 shows that high P (and $HCO_3$, not
shown) does occur in both refreshing water ($Na_{ex} > 1$) and in salinizing water ($Na_{ex} < -1$), but mainly when the $SO_4/Cl$ ratio is
below 0.14. Therefore, we infer that sulfate reduction/organic matter decomposition is the prime process in releasing P, and
is more discriminating high P than cation exchange processes. There is a high probability for sulfate reduction dominated
polder catchments to have very high $HCO_3$ concentration in groundwater according to Eq. [1]. In our study area, high $HCO_3$
concentration levels in both groundwater and surface water were mainly present in areas with marine sediments that contain
shell fragments and organic matter. The base level groundwater alkalinity from the dissolution of shell fragments and
carbonate minerals is further increased by the organic matter decomposition in the subsurface. This observation confirms the
earlier findings of Griffioen et al. (2013) who highlighted the relation between the nutrient concentrations and $pCO_2$ in these
marine sediments. The main chemical reactions involved are listed in Table 3.
The seepage of the alkalized groundwater increases alkalinity of the surface water, which is indicated by the high correlation
between groundwater and surface water $HCO_3$, and with Ca in surface water (Table 1). Subsurface organic matter
mineralization by processes like sulfate reduction and methanogenesis (Chapelle et al., 1987; Griffioen et al., 2013) (Table 3
[2], [6]), is a  probable major reason for  enhanced surface water $HCO_3$ in polders with brackish groundwater, like the
polders in the Zuiderzee margin and the Upconing polders (Fig. 8).
In the urban area of Amsterdam sand suppletion, which varies greatly in thickness and chemical composition, is another
source of alkalinity. Some of the sands contain shell fragments because of their marine origin. However, little is known
about the distribution of these calcite-rich sands. The poorly registered spatial distribution and sources of the supplied
calcite-rich sands might complicate the assessment of their impact on urban polder water quality.
Sulfate concentrations are higher in the receiving surface water than in the groundwater. We ascribe the sulfate surpluses
(Fig. 7) to additional sources affecting the surface water, including atmospheric deposition, agricultural inputs, sewer
leakage (Ellis, et al., 2005), storm runoff, and/or the oxidation of pyrite ($FeS_2$). Pyrite is ubiquitously present in this area
(Griffioen et al., 2013) and oxidizes in the topsoil, where either $O_2$ or $NO_3$ can act as electron acceptor (Wallenburg, 1975).
We suggest that sulfate concentrations are especially high in polders where shallow groundwater flow is enhanced by the
presence of tile drains in clay rich polders that needed this drainage system to prevent water tables rising into the root zone in



wet periods. Tile drain flow can bring the released $SO_4$ to the surface water. For urban polders with high $SO_4$ concentrations,
like the Zuiderzee margin region polders, sewer system leakage may be an additional source of $SO_4$. Aging and faulty
connections of pipes may result in a leakage of water with high $SO_4$ and nutrient concentrations.

**4.2 Groundwater contribution to surface water composition**

The groundwater in the upper 50 m of the subsurface of the study area is an important source of nutrients in the study area's
surface waters (Delsman, 2015). Brackish groundwater especially seeps up into the polders of the Zuiderzee margin region
and into the Upconing area. The seepage of paleo-marine, brackish groundwater is driven by the low surface water levels
after the lake reclamation and the drainage via pumping stations. De Louw et al. (2010) reported that this groundwater
seepage predominantly takes place via concentrated boils through the clay and peat cover layer.
The excess water in the Upconing area is re-used as inlet water for several downstream polder catchments, which extends the
impact of the brackish, alkaline, and nutrient rich groundwater to a larger scale. The water redistribution disturbs the
'natural' surface water quality patterns and local groundwater impact in the receiving polders, such as polder Botshol. The
redistributed water largely infiltrates and returns with variable travel times via the groundwater system back towards the
deep upconing polders.

**4.3 Other sources of nutrients**

Besides the contribution from nutrient rich groundwater seepage, this study indicated that there are other possible sources of
nutrients in the study area. In agricultural lands, a part of the applied nutrients is typically lost towards the surface water via
drainage and runoff. The high groundwater $NO_3$ concentrations in the Ice pushed ridge are caused by the infiltration of
agricultural water (Schot et al., 1992b). The high nitrate loads and concentrations in surface water and groundwater of the
polders in the southeast (e.g. # 122 (Muyeveld), # 140 ('t Gooi)) originate from agricultural activities in surrounding polders
In the urban polders within the Amsterdam city that have no significant seepage (average seepage ≤ 0), TP and TN EQSs are
frequently exceeded because of intensive human activities such as application of fertilizer, feeding ducks and fish, and point
emissions like sewer overflow leakage from the sewer system (pers. Comm. Waternet).
In the study area, the most intensively urbanized polders are mainly infiltrating and are more affected by inlet water
containing high Cl and $HCO_3$ concentrations than by groundwater. For deep urban polders, the situation is different. In these
polders, the influences of typical urbanization related water quality issues are masked by the large impact of brackish,
nutrient rich groundwater exfiltration. Although the paved area percentage in this paper was used as a variable representing
urban land cover influences, it seems not be the dominant landscape characteristic that governs the spatial patterns in polder
surface water quality. Urban water quality is determined by multiple factors, as was also concluded by several other studies
(Göbel et al., 2007; Vermonden et al., 2009). However, a better measurement method or classification of paved area
percentage may improve the explanatory power of this variable (Brabec et al., 2002).



The Vecht lakes polders with high surface water area percentages, representing lakes that are mainly used for recreation purposes, showed relatively low solute concentrations and loads in surface water (Fig. 10 and Fig. 11). In our study area, many lakes and polders with large surface water areas show large infiltration rates due to their elevation relative to other polders (Vermaat et al., 2010). Moreover, some of these lakes are replenished by inlet water that has passed a phosphate purification unit. In addition, the large open water area retains nutrient transport due to long residence times and ample opportunities for chemical and biological transformation processes like denitrification, adsorption, and plant uptake.

### 4.4 Uncertainties and perspectives

Due to the disturbance of urban constructions, combined with redistribution of water through artificial drainage corridors, water flow in lowland urban areas is more complex than in rural or non-low-lying and freely draining catchments. Natural patterns of water chemistry might be significantly disturbed and hydrochemical processes are masked. The understanding of urban water quality patterns might improve if the monitoring program would be extended with tracers that are typical for specific sources, such as sewage leakage or urban runoff. Most solutes that are currently measured can originate from various anthropogenic and natural sources.

In the statistical analysis, for each pair of variables, only polders with complete data were taken into account, which could result in a loss of information. Seepage data was simulated by a group of models of which the results may deviate from the hard to measure actual seepage. We used averages of groundwater concentrations and soil properties, which caused a loss of information on the spatial variation within the polders. The interpolation of groundwater quality also added uncertainty, for example hidden correlations for groundwater parameters. In addition, differences in sampling methods and analytical procedures between groundwater and surface water quality monitoring programs may add uncertainties. These uncertainties may all have influenced the data characteristics apart from the uncertainties in the concentration measurements caused by the sampling, transport, and analytical procedures.

In future studies, urban lowland catchments with and without seepage could be studied separately and more detailed land use or paved area categories could be included. The drainage and/or leakage from sewage systems and the drainage via tube drains should be taken into consideration. Drainage systems can provide a short-cut for solute transport towards surface water (Rozemeijer and Broers, 2007), leading to higher solute concentrations in surface water. High groundwater levels may induce groundwater discharge via the sewage or drainage systems (Ellis, 2005). In addition, studying the temporal variation of surface water quality will give more insights into how the groundwater impact on surface water quality functions, as well as on solutes transport and pathways in urban hydrological systems. A detailed monitoring network in several urban polder catchments, which is anticipated as further work, could yield a more complete insight into water and contaminant flow routes and their effects on surface water solute concentrations and loads.





## 5 Conclusion

In this paper, a clear groundwater impact on surface water quality was identified for the greater Amsterdam area. It was concluded that this groundwater seepage significantly impacts surface water quality in the polder catchments by introducing brackish, alkaline, and nutrient rich water. In general, nutrient concentrations in groundwater were much higher than in surface water and often exceeded surface water Environmental Quality Standards (EQSs) (in 93 % of the polders with available data for TP and in 91 % for TN) which indicates that groundwater is a large potential source of nutrients in surface water. Our results strongly suggest that organic matter mineralization is a major source of nutrients in the subsurface of coastal peat land areas. High correlations ($R^2$ up to 0.88) between solutes in groundwater and surface water confirmed the effects of surface water-groundwater interaction on surface water quality. Especially in seepage polders, groundwater is a major source of Cl, $HCO_3$, Ca and the nutrients N and P, leading to general exceedances of EQS's for N and P in surface waters. Redistribution of these high nutrient seepage waters in dry periods seems to lead to EQS exceedances in adjacent boezem systems and in the receiving polders. Surface water quality in the Amsterdam urban area is also influenced by groundwater seepage, but other anthropogenic sources, such as leaking and overflowing sewers might amplify the eutrophication problems.

## Acknowledgements

This work was funded through a CSC scholarship (No.201309110088) and supported by the Strategic Research Funding of TNO and Deltares. We would like to thank the Waternet organization for making available their regional data on surface water quality and appreciate the contributions by Jos Beemster, Jan Willem Voort and Jasper Stroom.

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



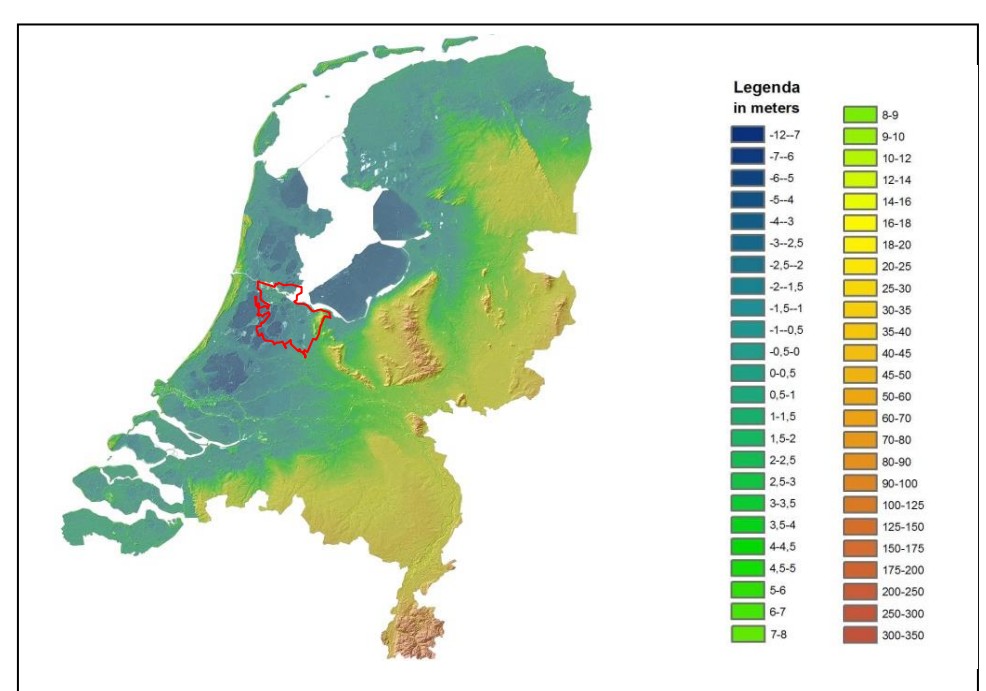

**Figure 1: Location of the research area (red) projected on the elevation map of The Netherlands (elevations in meters above mean sea level (MSL))**

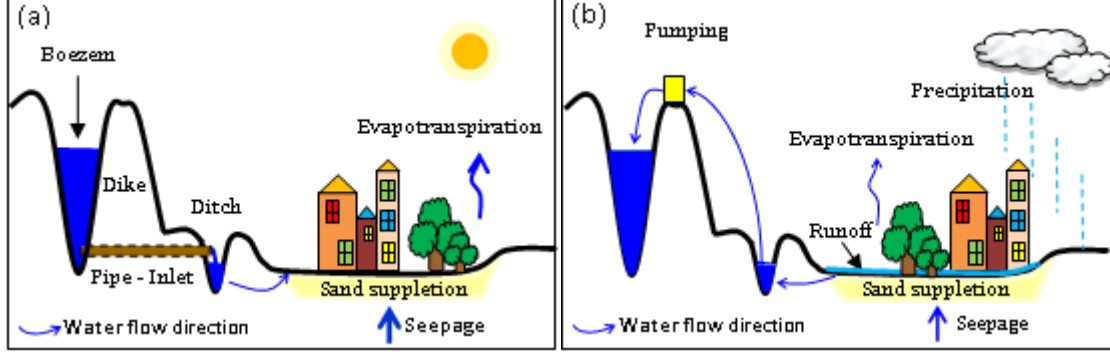

**Figure 2: Conceptual model of water fluxes in a polder system in times of water deficiency (a) and surplus (b).**



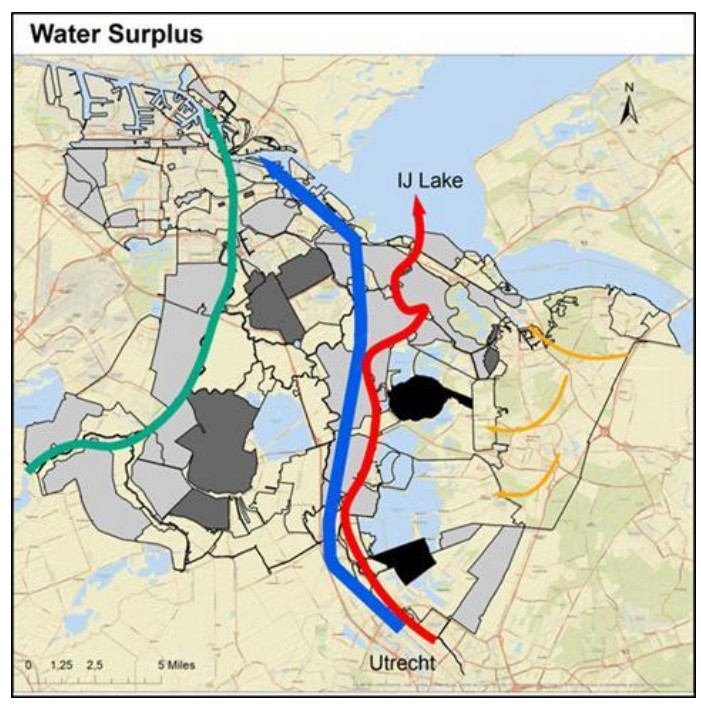

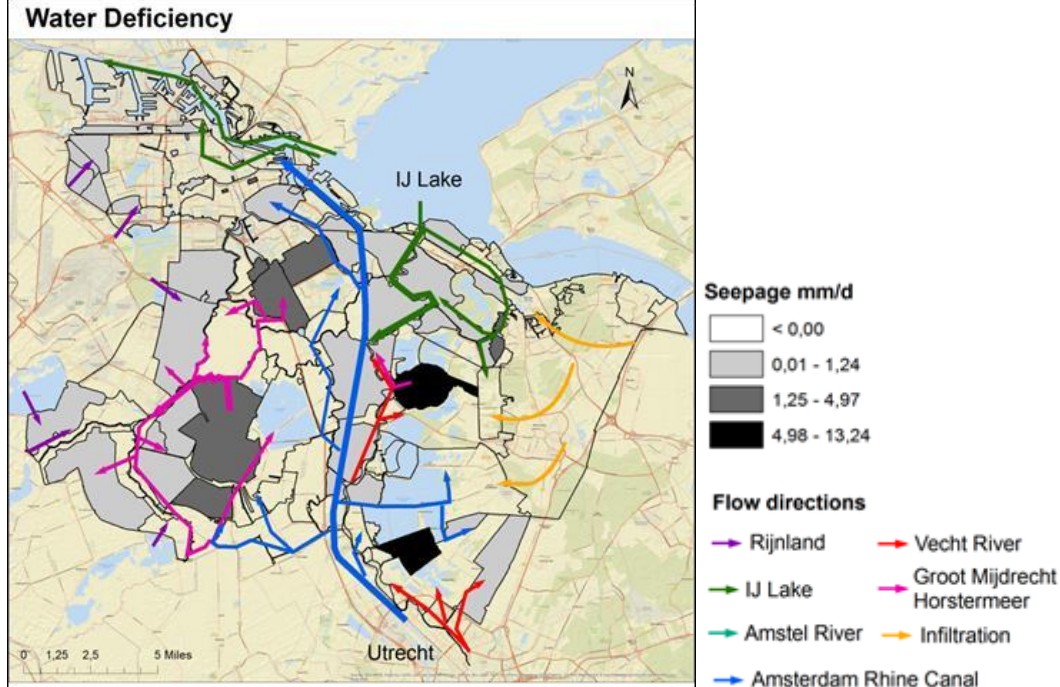

**Figure 3: Flow directions of surface water in water surplus and water deficiency period**





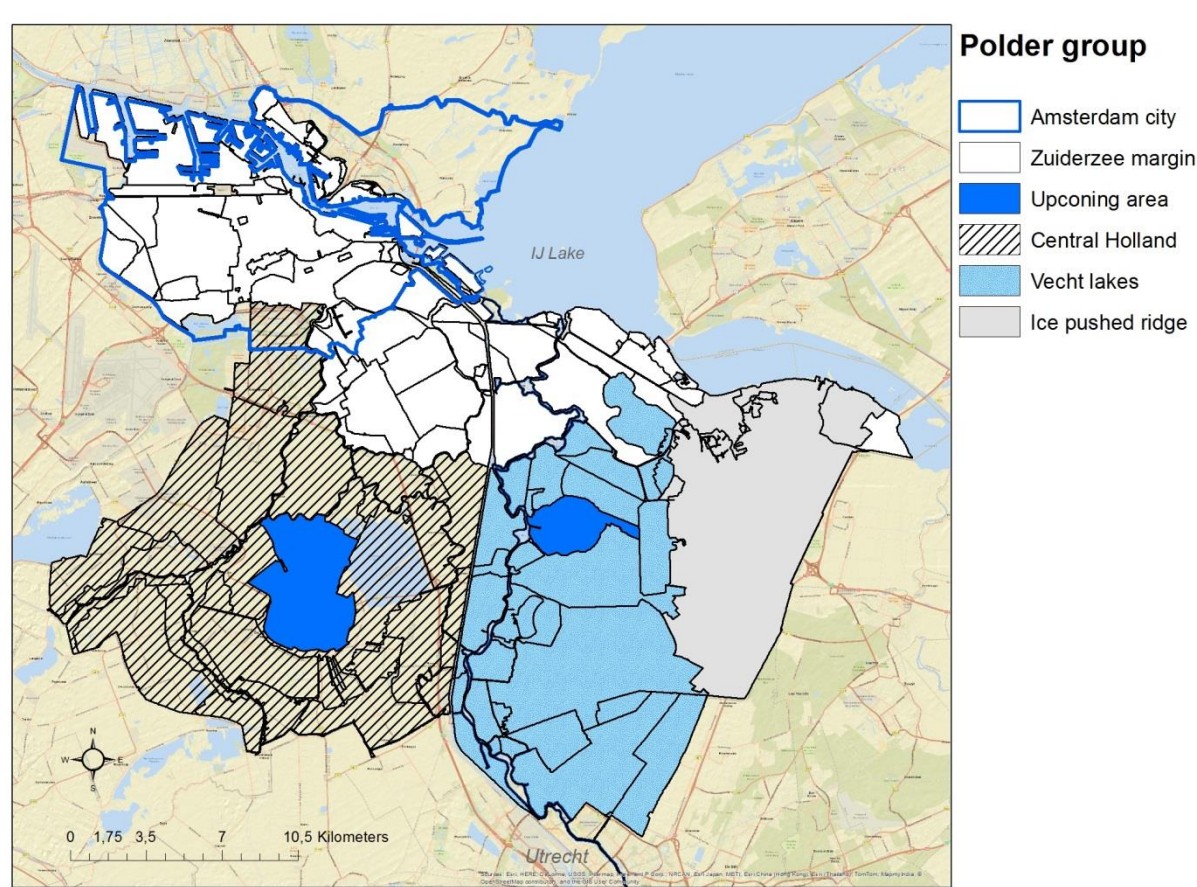

**Figure 4: Regions of the study area: (1) Zuiderzee margin, (2) Upconing area (deep brackish seepage polders Groot Mijdrecht and Horstermeer), (3) Central Holland, (4) Vecht lakes and 5) Ice pushed ridge. The Amsterdam city area is circled by the blue line.**





**Figure 5: Spatial variation of groundwater quality. 1-Zuiderzee margin, 2-Upconing area (Groot Mijdrecht and Horstermeer), 3-Central Holland, 4-Vecht lakes, 5-Ice pushed ridge (see Fig. 4). n is the amount of available data of each group. Boxplots show the distribution of solutes in the five regions. The two horizontal dashed lines for Cl indicate fresh water (<150 mg L$^{-1}$) and brackish water (>300 mg L$^{-1}$), respectively. Dashed lines represent EQSs for TN (2.4 mg L$^{-1}$) and TP (0.15 mg L$^{-1}$). The dashed line in the SO$_4$/Cl plot indicates the mass ratio of 0.14 in seawater (<0.14 indicates sulfate reduction; >0.14 indicates additional sources of sulfate besides (diluted) seawater)**





**Figure 6: Average groundwater concentrations (mg L⁻¹) per polder.**





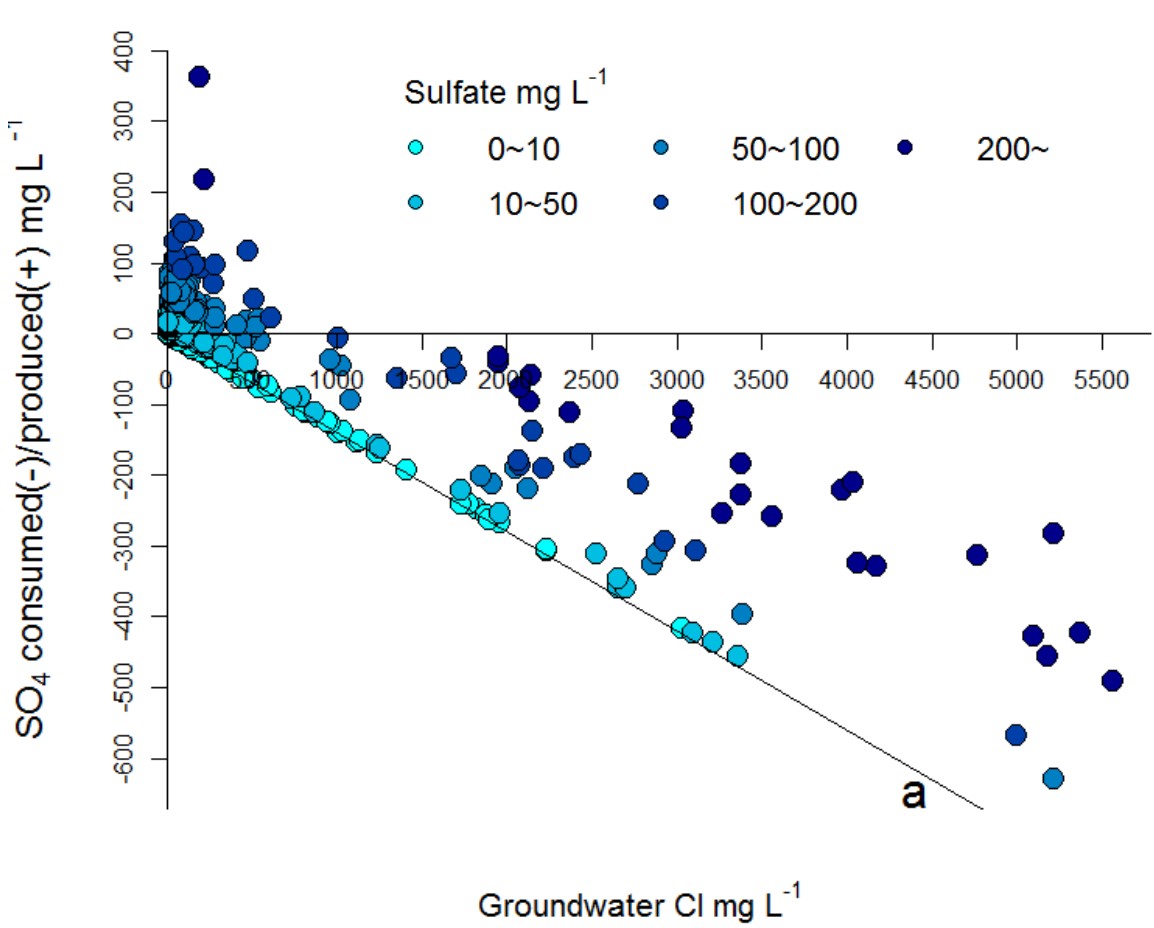

7 **Figure 7: Calculated concentration of sulfate reacted versus groundwater chloride concentration. The black line indicates the fresh**
8 **water – seawater mixing line where sulfate-reduction is complete.**





**Figure 8: Groundwater nutrient (TP and NH$_4$) concentrations with sulfate reduction (mass ratio SO$_4$/Cl, samples with value below 0.14 are considered to be affected by sulfate reduction and above 0.14 indicates sulfate production by natural or artificial processes). The symbols in (A) and (B) are colored by HCO$_3$ concentration and in (C) by Cl concentration.**





**Figure 9: Spatial variation of surface water quality. 1-Zuiderzee margin, 2-Upconing area, 3-Central Holland, 4-Vecht lakes (5-Ice**
**pushed ridge not shown due to insufficient data). n is the observation number of each group. The two horizontal dashed lines for**
**Cl indicate fresh water (<150 mg L$^{-1}$) and brackish water (>300 mg L$^{-1}$), respectively. Dashed lines in TP and TN represent EQSs**
**for TN (2.4 mg L$^{-1}$) and TP (0.15 mg L$^{-1}$). The dashed line in the SO$_4$/Cl plot indicates the mass ratio of 0.14 in seawater (<0.14**
**indicates sulfate reduction; >0.14 indicates additional sources of sulfate besides (diluted) seawater).**





**Figure 10:  Discharge-normalized average concentrations (mg L⁻¹) per polder**





4    **Figure 11: Surface water solute loads (average of 2010 to 2013) distribution maps in kg ha$^{-1}$y$^{-1}$**



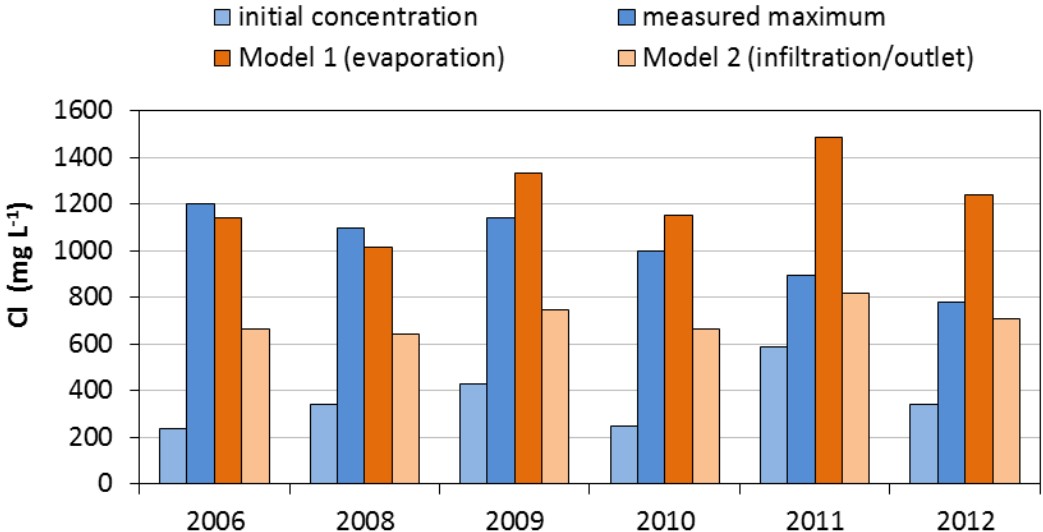

**Figure 12: Summary of the water and chloride balance for polder Botshol; the graph shows (1) the initial Cl before the water inlet season (light blue), (2) the resulting Cl peak in Botshol after some months of inlet (dark blue), and (3) the results of the two models (model 1 dark orange, model 2 light orange).**



**Table 1 Coefficients of determination between groundwater quality and surface water quality**

| | TP GW | TN GW | NH$_4$ GW | NO$_3$ GW | HCO$_3$ GW | SO$_4$ GW | Ca GW | Cl GW | TP SW | TN SW | NH$_4$ SW | NO$_3$ SW | HCO$_3$ SW | SO$_4$ SW | Ca SW | Cl SW |
|---|---|---|---|---|---|---|---|---|---|---|---|---|---|---|---|---|
| **TP GW** | | | | | | | | | | | | | | | | |
| **TN GW** | 0.65 | | | | | | | | | | | | | | | |
| **NH$_4$ GW** | 0.77 | 0.84 | | | | | | | | | | | | | | |
| **NO$_3$ GW** | | | | | | | | | | | | | | | | |
| **HCO$_3$ GW** | **0.68** | **0.63** | **0.82** | | | | | | | | | | | | | |
| **SO$_4$ GW** | -0.46 | | 0.41 | | | | | | | | | | | | | |
| **Ca GW** | | | | | 0.50 | | | | | | | | | | | |
| **Cl GW** | | | | | **0.48** | **0.40** | **0.77** | | | | | | | | | |
| **Paved area %** | | | | | | | | | | | | | | | | |
| **Elevation** | | | | | | | | | | -0.67 | -0.59 | -0.40 | -0.48 | -0.47 | -0.57 | |
| **Seepage rate** | | | | | | | | | | 0.48 | | 0.45 | | 0.46 | | |
| **Surface water %** | | | | | | | | | | | | | | | | |
| **Lutum %** | | | | | | | | | | | | | | | | |
| **Humus %** | | | | | 0.50 | | | | | 0.46 | 0.40 | | | | 0.41 | 0.48 |
| **Calcite %** | | | | | | | | | | | | | | | | |
| **TP SW** | 0.49 | 0.51 | 0.60 | | **0.64** | | | | | | | | | | | |
| **TN SW** | 0.45 | | 0.44 | | 0.52 | | | | 0.58 | | | | | | | |
| **NH$_4$ SW** | | | 0.44 | | 0.51 | | | | 0.49 | **0.77** | | | | | | |
| **NO$_3$ SW** | | | | | | | | | | 0.57 | | | | | | |
| **HCO$_3$ SW** | 0.57 | 0.55 | 0.64 | | **0.68** | | | 0.41 | **0.62** | 0.47 | 0.67 | | | | | |
| **SO$_4$ SW** | | | | | | | | | | 0.57 | | 0.50 | | | | |
| **Ca SW** | 0.59 | 0.54 | 0.63 | | **0.71** | | | 0.41 | **0.55** | 0.57 | 0.64 | | 0.88 | | | |
| **Cl SW** | | | | | 0.47 | | 0.47 | **0.69** | | 0.47 | **0.51** | | 0.52 | 0.49 | 0.55 | |

\* Only coefficients higher than or equal to 0.40 were shown in the table

   TP sw: surface water TP concentration in mg L$^{-1}$

   TP gw: groundwater TP concentration in mg L$^{-1}$



**Table 2 Linear regression results of each surface water solute (Spearman)**

| | $n_1$ | $n_2$ | $n_3$ | $n_4$ | $R^2$ | $R^2$ with only seeping polders | |
|---|---|---|---|---|---|---|---|
| $TP_{SW}$ | $+ HCO_{3\ GW}$ | $+ TP_{GW}$ | - Elevation | $-SO_{4\ GW}/Cl_{GW}$ | 0.42 | 0.51 | (9) |
| $TN_{SW}$ | $+ HCO_{3\ GW}$ | - Elevation | $+ P_{Humus}$ | $-SO_{4\ GW}/Cl_{GW}$ | 0.54 | 0.50 | (10) |
| $NH_{4\ SW}$ | $+ HCO_{3\ GW}$ | $+ NH_{4\ GW}$ | - Elevation | $-SO_{4\ GW}/Cl_{GW}$ | 0.41 | 0.58 | (11) |
| $NO_{3\ SW}$ | - Elevation | $- Cl_{GW}$ | $-SO_{4\ GW}$ | | 0.18 | 0.23 | (12) |
| $HCO_{3\ SW}$ | $+ HCO_{3\ GW}$ | + Seepage | | | 0.53 | 0.62 | (13) |
| $SO_{4\ SW}$ | - Elevation | | | | 0.21 | 0.20 | (14) |
| $Ca_{SW}$ | $+ HCO_{3\ GW}$ | - Elevation | | | 0.59 | 0.59 | (15) |
| $Cl_{SW}$ | $+ HCO_{3\ GW}$ | $+ Cl_{GW}$ | - Elevation | | 0.54 | 0.54 | (16) |

\* '+' positive relation, '-' negative relation

$n_1$: first variable, the most significant variable

$TP_{SW}$: surface water TP concentration in mg L$^{-1}$

$TP_{GW}$: groundwater TP concentration in mg L$^{-1}$

Elevation: average polder elevation in m N.A.P

Seepage: seepage rate in mm y$^{-1}$

$P_{Humus}$: percentage of humus in the soil profile sample

**Table 3 Main hydrogeochemical reactions in the study area**

| Process | | Reactions | No |
|---|---|---|---|
| Organic matter decomposition | | $CH_2O\ N_xP_y \rightarrow xN + yP + HCO_3^- +$ other components | [2] |
| | | $CH_2O\ N_xP_y + O_2 \rightarrow CO_2 + H_2O + xN + yP$ | [3] |
| | | $5CH_2O\ N_xP_y + 4NO_3^- \rightarrow 2N_2 + CO_2 + 4HCO_3^- + 3H_2O + 5xN + 5yP$ | [4] |
| | | $2CH_2O\ N_xP_y + SO_4^{2-} \rightarrow H_2S + 2HCO_3^- + 2xN + 2yP$ | [5] |
| Pyrite oxidation | | $2CH_2O\ N_xP_y \rightarrow CH_4 + CO_2 + 2xN + 2yP$ | [6] |
| | | $2FeS_2 + 7O_2 + 2H_2O \rightarrow 2Fe^{2+} + 4SO_4^{2-} + 4H^+$ | [7] |
| | | $5FeS_2 + 14NO_3^- + 4H^+ \rightarrow 5Fe^{2+} + 10SO_4^{2-} + 7N_2 + 2H_2O$ | [8] |
| Calcite dissolution | Closed system | $CaCO_3 + H_2O \leftrightarrow Ca^{2+} + HCO_3^- + OH^-$ | [9] |
| | Open system | $CaCO_3 + CO_2 + H_2O \leftrightarrow Ca^{2+} + 2HCO_3^-$ | [10] |