# Peer review of "Groundwater impacts on surface water quality and nutrient loads in lowland polder catchments: monitoring the greater Amsterdam area"

_Hydrology and Earth System Sciences, 2017_

## Referee Comment (RC1) · Anonymous Referee #1 · 28 Apr 2017

The paper by YU et al. presents analysis of water quality conditions and nutrient chemistry in polders found in the Amsterdam area. I found the analysis to be well done and I have no issues with it.

My main criticism of the paper is the lack of context for any readers outside of the Amsterdam area. The authors immediately jump into the specifics of the Amsterdam/Netherlands region and describe the area in great detail. Many times, the paper reads like a history of the region and monitoring activities conducted in the area. Those reading the paper from anywhere other than the Netherlands are left wondering why they should care about it. The authors need to spend some time introducing and discussing the larger issues relevant to the rest of the world. How are the issues and

study methods and results used in the paper relevant to other locations and issues? This is a matter of doing some homework to see what has been done related to these systems or similar environments.

This is my only comment but it is not trivial. In my opinion, the authors need to provide some global relevance for this analysis before it can be accepted. The authors need to make a case for why anyone outside of the Netherlands area should care about this topic and their results.

---

## Referee Comment (RC2) · Anonymous Referee #2 · 20 Aug 2017

HESS Research Article

hess-2017-99

Groundwater impacts on surface water quality and nutrient loads in lowland polder catchments: Monitoring the greater Amsterdam area

In the present work, the authors attempt to characterize the impacts of groundwater seepage on the polder network around Amsterdam by exploiting data from the dense network of groundwater and surface water monitoring in this area. The authors combine water quality monitoring data with other biophysical characteristics of 144 polders and take a statistical approach to bettering our understanding of sources, transport mechanisms, and pathways in this area. They conclude that groundwater is a major source of nutrients in this mixed urban/agricultural catchment. In particular, they note that elevated nutrient and bicarbonate concentrations in the groundwater seepage originate from decomposition of organic matter in subsurface sediments coupled to sulfate reduction and possibly methanogenesis. Their results suggest that groundwater-surface water interactions are important to nutrient dynamics in urbanizing delta regions.

The current work is important, as it attempts to tease out the relative importance of natural and anthropogenic sources of nutrients within the region and to elucidate why implementation of nutrient management practices may not effectively reduce surface water concentrations to target levels, particularly in urban areas. The approach used in the paper, which combines correlation analysis between surface water and groundwater quality, as well as statistical analysis of relationships between landscape characteristics provides an interesting perspective on the drivers of various solute concentrations in surface water.

The study does, however, leave some questions unanswered. First, in the abstract it is claimed that "land use" is used as a variable in the multiple linear regression, which attempts to identify the strongest drivers of surface water nutrient concentrations. In the analysis, however, the only land-use variable that I see is "paved area." As the authors mention more than once that agriculture in the polder catchments could be driving surface water nutrient concentrations (and I would agree), I find it puzzling that this is not used as a potential variable for the regression analysis. Second, the authors average groundwater data taken over a period of more than 100 years, but do not discuss how groundwater levels may have change over time, and how these trajectories may have differed from place to place, thus affecting use of the GW data in the spatial analysis. Finally, it is unclear how issues of collinearity impact the results of the correlation analysis and development of the multiple linear regression model. A more complete treatment and subsequent discussion of possible collinearity between independent variables would strengthen the analysis.

**Specific comments:**

p. 6, ll. 8-12     You describe here the variables used in your analysis, but do not include any land-use variable other than "paved area." Clearly, agricultural area is a major factor

driving concentrations in your study area, so it seems a large omission to not include it in your analysis. Is it simply that the agricultural area was not included in the database that you utilized? If so, could you obtain that information through other sources of land-use data? It is possible that including agricultural area in your analysis would significantly change the findings of your analysis regarding significant drivers of surface water concentrations.

p. 6, ll. 14-18    In your methods, you mention that for each well, you average concentrations for each monitoring well (at individual monitoring screens) for all sampling dates. You also mention that the groundwater data is from the period 1910-2013—more than 100 years. I would assume that there could have been significant changes in groundwater quality over that period, and that the temporal patterns of change could have differed across the study period. Accordingly, is it correct to combine all sampling data across this 100-year period, or in doing so are you conflating spatial and temporal differences across the study area?

p. 8, ll. 20-30    You do not discuss here how you dealt with issues of collinearity among the explanatory variables. For example, there are clearly high correlations (r>0.60) among some of the groundwater solute concentrations (particularly with regard to $HCO_3$). With this being the case, how do you (from a quantitative perspective) make decisions regarding inclusion in the multiple linear regression model? For example, in your MLR equation for TP, you include both $HCO_{3(GW)}$ and $TP_{(GW)}$, although your correlation table in Table 1 shows a reasonably high collinearity (r=0.68) between these two variables. How do you justify use of both of them in the MLR equation?

p. 12, ll. 22-23    You say here that ammoniums correlates more strongly with TN than nitrate and conclude that ammonium is therefore likely the main form of TN in the study area. When I look at Fig. 5, however, it appears that nitrate is likely the dominant form of N in the ice-pushed ridge area (5) and possibly the Vecht Lakes area (4). It might be more useful to discuss the actual variations among locations (and reasons why), rather than just to cite the simple regression results.

p. 13, ll. 8-21    You discuss the results of the MLR analysis here, but do not reference the table that contains the results. Please include the table reference here.

Fig. 5    It is very difficult to understand the variations in concentrations of solutes among locations in these figures due to the different concentration ranges from site to site. For example, for TN, all of the concentration ranges look very similar, simply because you scale the y-axis to include all of the outlier values for site #5. Is it important to include all of the outliers? I would recommend plotting these in such a way that you allow the reader to understand differences in median and interquartile range values, rather than prioritizing the representation of outliers.

Table 1    For your correlation analysis, you should include the 1.0 values to show perfect correlation between two identical variables. This will help add structure to the table and make it easier to understand.

---

## Author Comment (AC1) · 15 Sep 2017

We thank the reviewer for his/her review of our paper which helped to improve the manuscript. We thoroughly worked on the comment, and the responses are given below.

**General comments**

*The paper by YU et al. presents analysis of water quality conditions and nutrient chemistry in polders found in the Amsterdam area. I found the analysis to be well done and I have no issues with it.*

*My main criticism of the paper is the lack of context for any readers outside of the Amsterdam area. The authors immediately jump into the specifics of the Amsterdam/Netherlands region and describe the area in great detail. Many times, the paper reads like a history of the region and monitoring activities conducted in the area. Those reading the paper from anywhere other than the Netherlands are left wondering why they should care about it. The authors need to spend some time introducing and discussing the larger issues relevant to the rest of the world. How are the issues and study methods and results used in the paper relevant to other locations and issues? This is a matter of doing some homework to see what has been done related to these systems or similar environments.*

*This is my only comment but it is not trivial. In my opinion, the authors need to provide some global relevance for this analysis before it can be accepted. The authors need to make a case for why anyone outside of the Netherlands area should care about this topic and their results.*

Agreed. We thank the reviewer for the compliments and we agree with the comment that we should elaborate more on the relevance of our study for other areas outside Amsterdam and The Netherlands. Although we mentioned in the abstract that "we expect that taking account of groundwater-surface water interaction is also important in other subsiding and urbanising deltas around the world, where water is managed intensively in order to enable agricultural productivity and achieve water sustainable cities.", we agree that we did not make this very specific in the rest of the manuscript. We intend to add text on this issue in the revised paper. We will elaborate on the international relevance of the paper in the introduction, the discussion and the conclusions part of the paper.

In the introduction, we will add the following text after the first sentence:

> Lowland deltas account for 2 % of the world's land, but accommodated around 600 million people in 2000, and about 1400 million by 2060 as was estimated by Neumann et al. (2015) [1]. The reclamation of swamps and lakes and the drainage of peat areas to enable urbanisation and agriculture severely change the hydrological, chemical and ecological environment of these areas (Ellis et al., 2005; Yan et al., 2017) [2,3]. Lowland delta areas are vulnerable for water
* * *
[1] B. Neumann, A.T. Vafeidis, J. Zimmermann, R. J. Nicholls: Future Coastal Population Growth and Exposure to Sea-Level Rise and Coastal Flooding - A Global Assessment, Plos One, 10, 3, 2015. e0118571. doi:10.1371/journal.pone.0118571.

[2] J. B. Ellis, J. Marsalek, B. Chocat: Encyclopedia of Hydrological Sciences, Urban Water Quality, 1st edition, M G Anderson, John Wiley & Sons, Ltd, United States, 8, 97, 2005.

quality deterioration by processes like salinization and eutrophication, which can be amplified by climate change (Wu et al., 2015)[4] and land subsidence (Minderhoud et al., 2017)[5].

In the discussion section, our study indicates that groundwater seepage can be a significant and even dominating source of nutrients in lowland areas where water is pumped out of polder systems that without pumping would turn into fresh water lakes. Our study shows that the groundwater seepage leads eutrophication and that redistributing water out of some deep elevation polders with upconing brackish water had further spread the nutrients to the whole water system. We will add the following paragraphs:

*Section 4.2 Similar patterns are expected to be present in other lowland areas, which are highly manipulated by human. Typical delta areas where subsurface processes are expected to release nutrients from reactive organic matter and peat in the subsurface are the Mekong delta (Minderhoud et al., 2017), the Mississippi delta (Törnqvist et al., 2008)[6], and the Sacramento-San Joaquin delta (Drexler et al., 2009)[7]. In many of these areas the water management shows resemblance to the Dutch situation. However, the large amount of groundwater quality and surface water quality data that was available in our study area is unique. Still, signals of groundwater influence on nutrient concentrations were reported from eastern England (pers. comm. M.E. Stuart, British Geological Survey) and from the lowland parts of Denmark (Kronvang et al. 2013)[8].*

*Section 4.3 line 20-21: However, the results indicates that reducing the amounts of manure and fertilizer and the associated N and P inputs in agriculture might not contribute enough in reducing N and P concentrations and fluxes for environmental purposes, as the N and P concentrations in the surface water are dominantly caused by seepage of groundwater. This certainly holds for urban areas where these inputs are absent (see new Figure Supplementary Info). Given the large loads of N and P that originate from one large polder with upconing brackish groundwater - the Groot Mijdrecht polder - one of the solutions proposed in The Netherlands was to turn this area back into a fresh water lake. By doing so, the seepage of nutrient rich groundwater would stop as the higher water levels would lead to neutral or even infiltrating conditions. However, this proposal led to a lot of protest among the municipalities and farming communities in the polder and was not considered feasible given the economic values that were involved. This example shows that the reclamation of swamps and lakes for*
* * *
[3] R. Yan, J. Huang, L. Li, J. Gao: Hydrology and phosphorus transport simulation in a lowland polder by a coupled modeling system, Environ Pollut, 227, 613-625, 2017.

[4] J. Wu, M. E. Malmstrom: Nutrient loadings from urban catchments under climate change scenarios: Case studies in Stockholm, Sweden, Sci Total Environ, 518-519, 393-406, 2015.

[5] P. S. J. Minderhoud, G. Erkens, V. H. Pham, V. T. Bui, L. Erban, H. Kooi, E. Stouthamer: Impacts of 25 years of groundwater extraction on subsidence in the Mekong delta, Vietnam, Environ Res Lett., 12, 2017.

[6] T. E. Törnqvist, D. J. Wallac, J. E. A. Storms, J. Wallinga, R. L. Van Dam, M. Blaauw, M. S. Derksen, C. J. W. Klerks, C. Meijneken, E. M. A. Snijders: Mississippi Delta subsidence primarily caused by compaction of Holocene strata. Nat Geosci, 1, 3, 173-176, 2008.

[7] J. Z. Drexler, C. S. De Fontaine, S. J. Deverel: The legacy of wetland drainage on the remaining peat in the Sacramento San Joaquin Delta, California, USA, Wetlands, 29,1, 372-386, 2009.

[8] B. Kronvang, J. Køgestrand, J. Windolf, N. Ovesen, L. Troldborg: Background phosphorus concentrations in Danish groundwater and surface water bodies, EGU General Assembly 2013, 7-12 April, 2013, Vienna, Austria, id. EGU2013-2249.

*urbanisation or agriculture can lead increased nutrient loads to surface waters in the surroundings which are hard to mitigate. This scenario has wider implications for water management in other urbanising lowland areas around the world.*

In the conclusion, we will add the following sentences:

*Our results strongly suggest that organic matter mineralization is a major source of nutrients in lowland deltas where water levels are lowered to enable urbanisation and agricultural land use. The discharge and redistribution of nutrient rich water from reclaimed lakes and swamps enhances eutrophication in downstream water resources and is hard to mitigate.*

---

## Author Comment (AC2) · 16 Sep 2017

We thank the reviewer for his/her review of our paper which helped to improve the manuscript. We thoroughly worked on the comments, and the responses are given below.

**General comments**

*In the present work, the authors attempt to characterize the impacts of groundwater seepage on the polder network around Amsterdam by exploiting data from the dense network of groundwater and surface water monitoring in this area. The authors combine water quality monitoring data with other biophysical characteristics of 144 polders and take a statistical approach to bettering our understanding of sources, transport mechanisms, and pathways in this area. They conclude that groundwater is a major source of nutrients in this mixed urban/agricultural catchment. In particular, they note that elevated nutrient and bicarbonate concentrations in the groundwater seepage originate from decomposition of organic matter in subsurface sediments coupled to sulfate reduction and possibly methanogenesis. Their results suggest that groundwater-surface water interactions are important to nutrient dynamics in urbanizing delta regions.*

*The current work is important, as it attempts to tease out the relative importance of natural and anthropogenic sources of nutrients within the region and to elucidate why implementation of nutrient management practices may not effectively reduce surface water concentrations to target levels, particularly in urban areas. The approach used in the paper, which combines correlation analysis between surface water and groundwater quality, as well as statistical analysis of relationships between landscape characteristics provides an interesting perspective on the drivers of various solute concentrations in surface water.*

We acknowledge the reviewer for his thorough review and the positive words about our manuscript.

*The study does, however, leave some questions unanswered. First, in the abstract it is claimed that "land use" is used as a variable in the multiple linear regression, which attempts to identify the strongest drivers of surface water nutrient concentrations. In the analysis, however, the only land-use variable that I see is "paved area." As the authors mention more than once that agriculture in the polder catchments could be driving surface water nutrient concentrations (and I would agree), I find it puzzling that this is not used as a potential variable for the regression analysis.*

Agreed. The reviewer is correct that we did not explicitly use agricultural land use and inputs in our analysis. Implicitly, there is land use in the analysis already, because both the soil types and elevation variables distinguish between dairy farming (shallow water level peat polders with somewhat higher elevation) and arable farming (deeper clay polders with deeper ditches). The higher N and P loads in low elevation polders may be partly caused by more intensive arable land use. Evaluating this major comment of the reviewer, we acknowledge that paved area percentage, elevation and soil types are very poor measures of agricultural practices, and we therefore now included the application rates of manure and fertilizer in the analysis which are relatively well known in The Netherlands because of

the advanced bookkeeping system of farmers and the central registration. In the revised paper, we will describe the newly introduced data (N and P inputs in kg/ha/y), compare the N and P inputs with surface water annual loads, show the spatial distribution of N and P inputs in maps in the Supplementary Info, which allows the reader to visually compare the nutrient inputs spatial pattern with the pattern of surface water loads.

The newly introduced data include the annual animal manure and fertilizer inputs of N and P. These new data have been retrieved from the central farmer bookkeeping data for nutrient fate and transport model calculations using INITIATOR (Wolf et al. 2003)[1]. The N and P input data are valid for the year 2011, which corresponds with most of our surface water data.

The determination coefficients ($R^2$) of the new statistical analysis including the 2 new land use variables are shown in the table below (Table 1). All determination coefficients for the newly included variables are remain below the threshold of 0.40 (Table 1), but a slight negative correlation was found between the N inputs in kg/ha/y and the normalized concentrations of HCO3 in surface water and the total N concentrations of groundwater (range -0.30 ~ -0.33).

This analysis confirms our initial assumption that the inputs of N and P from agriculture are not the major factors determining the N and P concentrations in surface water and groundwater in this area. The additional analysis strongly confirms our earlier findings and will help us to even better describe the large influence of groundwater seepage, the related subsurface geochemical processes that define them and the subsequent redistribution of water through these polder systems on surface water chemistry and nutrient concentrations. This is interesting to other readers, as the lowlands in the western Netherlands around Amsterdam are still part of one of the most intensive agricultural regions worldwide, and unraveling nutrient problems in this region can help to understand other lowland regions better. Naturally, we will include this analysis in the final revised paper and thank the reviewer for his/her comment that we think really helped to emphasize our conclusions and certainly will improve the manuscript.

Using the newly introduced variables, we updated Table 1 of the original paper. We intend to add 2 extra maps to the Supplementary Info of the paper (see below), showing the distribution of the N and P inputs over the greater Amsterdam regions, which enable the visual comparison of the N and P patterns in groundwater and surface water, with the inputs from agriculture. The visual comparison confirms the statistical conclusions that N and P input patterns don't match the N and P concentrations and load maps. Especially in the urban areas of the city of Amsterdam, high concentrations of N and P appear in groundwater and surface water whereas the agricultural inputs in those areas are minimal. These results will be discussed in the discussion part of the paper.                         .

[1] J. Wolf, A.H.W. Beusen, P. Groenendijk, T. Kroon, R. Rotter, H. van Zeijts: The integrated modeling system STONE for calculating nutrient emissions from agriculture in the Netherlands, Environ Modell & Softw, 18: 597–617, 2003.

Revised **Table 1 Coefficients of determination between groundwater quality and surface water quality**

| | TP GW | TN GW | NH$_4$ GW | NO$_3$ GW | HCO$_3$ GW | SO$_4$ GW | Ca GW | Cl GW | TP SW | TN SW | NH$_4$ SW | NO$_3$ SW | HCO$_3$ SW | SO$_4$ SW | Ca SW | Cl SW |
|---|---|---|---|---|---|---|---|---|---|---|---|---|---|---|---|---|
| TP $_{GW}$ | 1 | | | | | | | | | | | | | | | |
| TN $_{GW}$ | 0,65 | 1 | | | | | | | | | | | | | | |
| NH$_4$ $_{GW}$ | 0,77 | 0,84 | 1 | | | | | | | | | | | | | |
| NO$_3$ $_{GW}$ | | | | 1 | | | | | | | | | | | | |
| HCO$_3$ $_{GW}$ | 0,68 | 0,63 | 0,82 | | 1 | | | | | | | | | | | |
| SO$_4$ $_{GW}$ | -0,46 | | | 0,41 | | 1 | | | | | | | | | | |
| Ca $_{GW}$ | | | | | 0,50 | | 1 | | | | | | | | | |
| Cl $_{GW}$ | | | | | 0,48 | 0,40 | 0,77 | 1 | | | | | | | | |
| TP $_{SW}$ | 0,49 | 0,51 | 0,60 | | 0,64 | | | | 1 | | | | | | | |
| TN $_{SW}$ | 0,45 | | 0,44 | | 0,52 | | | | 0,58 | 1 | | | | | | |
| NH$_4$ $_{SW}$ | | | 0,44 | | 0,51 | | | | 0,49 | 0,77 | 1 | | | | | |
| NO$_3$ $_{SW}$ | | | | | | | | | | 0,57 | | 1 | | | | |
| HCO$_3$ $_{SW}$ | 0,57 | 0,55 | 0,64 | | 0,68 | | | 0,41 | 0,62 | 0,47 | 0,67 | | 1 | | | |
| SO$_4$ $_{SW}$ | | | | | | | | | | 0,57 | | 0,50 | | 1 | | |
| Ca $_{SW}$ | 0,59 | 0,54 | 0,63 | | 0,71 | | | 0,41 | 0,55 | 0,57 | 0,64 | | 0,88 | | 1 | |
| Cl $_{SW}$ | | | | | 0,47 | | 0,47 | 0,69 | | 0,47 | 0,51 | | 0,52 | 0,49 | 0,55 | 1 |
| N input kg ha$^{-1}$ y$^{-1}$ | | | | | | | | | | | | | | | | |
| P input kg ha$^{-1}$ y$^{-1}$ | | | | | | | | | | | | | | | | |
| Paved area % | | | | | | | | | | | | | | | | |
| Elevation | | | | | | | | | -0,67 | -0,59 | -0,40 | -0,48 | -0,47 | -0,57 | | |
| Seepage rate | | | | | | | | | | 0,48 | | | 0,45 | 0,46 | | |
| Surface water % | | | | | | | | | | | | | | | | |
| Lutum % | | | | | | | | | | | | | | | | |
| Humus % | | | | | 0,50 | | | | 0,46 | 0,40 | | | | | 0,41 | 0,48 |
| Calcite % | | | | | | | | | | | | | | | | |

\* Only coefficients higher than or equal to 0.40 were shown in the table

TP sw: surface water TP concentration in mg L$^{-1}$

TP gw: groundwater TP concentration in mg L$^{-1}$

[Figure]

Maps to be included in the Supplementary Info showing the spatial distribution of N and P inputs from agricultural land use.

*Second, the authors average groundwater data taken over a period of more than 100 years, but do not discuss how groundwater levels may have change over time, and how these trajectories may have differed from place to place, thus affecting use of the GW data in the spatial analysis.*

We agree with the reviewer that we should have discussed the use of groundwater quality data of such a long period of time. We will elaborate on that in the revised version of the paper. The underlying assumption in our analysis was that groundwater composition changes very slowly over time, and we wanted to use as much of the available groundwater data as possible to cover the entire region and all the polders studied with sufficient data to enable the statistical analysis. The large majority of the groundwater quality data we used is from the last 30 years (for example, 85% for chloride and 93% of P is sampled after 1980) and we do not expect that using the older data creates a significant bias to the results of the study, because hydrogeochemical processes in the reactive subsurface such as sulfate reduction and methanogenesis have a smoothing effect on the water composition in this area. Moreover, the overall flow patterns haven't changed much in the past 30 to 100 years, because the flow systems are completely determined by the water levels maintained in the polder systems which have not changed dramatically over the past 100 years. However, the interface between fresh and salt water is known to slowly move into the direction of a new equilibrium (Oude Essink et al. 2010)[2], but the process is known to be very slow and to continue over the next 200 years. We will further elaborate this issue in the revised version of the paper.

*Finally, it is unclear how issues of collinearity impact the results of the correlation analysis and development of the multiple linear regression model. A more complete treatment and subsequent discussion of possible collinearity between independent variables would strengthen the analysis.*

We agree with the reviewer that the method of the regression analysis is not well enough described in the paper. The variables to be integrated into multiple linear regression models for predicting surface water solute concentrations were selected based on the correlation matrix (Table 1). Again, the Spearman method was applied and linear regression was based on ranks in order to avoid outliers to determine the outcomes. The explaining variables for surface water concentrations include groundwater solute concentrations, landscape characteristics and the newly introduced N and P inputs from agriculture. We adopted the method described by Rozemeijer et al. (Rozemeijer et al., 2010)[3], who describe a form of
* * *
[2] G. H. P. Oude Essink, E. S. Van Baaren, P. G. B. De Louw: Effects of climate change on coastal groundwater systems: A modeling study in the Netherlands, Water Resour Res, 46, 10, 2010.

[3] J. C. Rozemeijer, Y. Van der Velde, F. Van Geer, G.H. de Rooij, P. J. J. F. Torfs, H. P. Broers: Improving load estimates of N and P in surface waters by characterizing the concentration response to rainfall events, Environ Sci & Technol, 44, 6305-6312, 2010.

sequential multiple regression analysis, where subsequently variables were added to the regression evaluating the accumulating effect on the resulting coefficient of determination $R^2$. The regression analysis started with a singular regression with the highest coefficient of determination ($R^2$) for explaining the surface water quality parameter under consideration. Subsequently, we searched for the best regression model with two and three explaining variables, where we accepted an additional variable only when the coefficient of determination $R^2$ increased by at least 3%. In this method, dependent variables can still add to the resulting $R^2$ as the coefficient of determination $R^2$ of the individual dependent variable pair is seldom larger than 0.7, pointing to different explaining power of the individual variables. Based on the reviewer's comments, however, we carefully scrutinized the regression results and including the two new explaining variables and found some regressions that led to improved $R^2$ in our analysis. The resulting regression table is reported below and will be described and discussed in the paper. For two surface water variables, we found the N inputs for agricultural led to a small but significant improvement of the explaining power of the regression.

**Revised table 2 Linear regression results of each surface water solute (Spearman)**

| | $n_1$ | $n_2$ | $n_3$ | $R^2$ | $R^2$ with only seeping polders | |
|---|---|---|---|---|---|---|
| $TP_{SW}$ | + $HCO_{3\,GW}$ | + $NH4_{GW}$ | | 0.43 | 0.49 | (9) |
| $TN_{SW}$ | - Elevation | + $HCO_{3\,GW}$ | + $N_{input}$ | 0.57 | 0.48 | (10) |
| $NH_{4\,SW}$ | - Elevation | + $HCO_{3\,GW}$ | + Seepage | 0.50 | 0.61 | (11) |
| $NO_{3\,SW}$ | - Elevation | + $N_{input}$ | | 0.18 | 0.23 | (12) |
| $HCO_{3\,SW}$ | + $HCO_{3\,GW}$ | + Seepage | + $NH4_{GW}$ | 0.57 | 0.70 | (13) |
| $SO_{4\,SW}$ | - Elevation | + $SO4_{GW}$ | | 0.25 | 0.22 | (14) |
| $Ca_{SW}$ | + $HCO_{3\,GW}$ | - Elevation | + Seepage | 0.65 | 0.63 | (15) |
| $Cl_{SW}$ | + $Cl_{GW}$ | + $HCO_{3\,GW}$ | + $P_{Humus}$ | 0.57 | 0.51 | (16) |

* '+' positive relation, '-' negative relation

$n_1$: first variable, the most significant variable

$HCO_{3\,SW}$: surface water $HCO_3$ concentration in mg $L^{-1}$

$HCO_{3\,GW}$: groundwater $HCO_3$ concentration in mg $L^{-1}$

Elevation: average polder elevation in m N.A.P

Seepage: seepage rate in mm $y^{-1}$

$P_{Humus}$: percentage of humus in the soil profile sample

$N_{input}$: manure and fertilizer N input in kg $ha^{-1} y^{-1}$

**Specific comments:**

*p. 6, ll. 8-12 You describe here the variables used in your analysis, but do not include any landuse variable other than "paved area." Clearly, agricultural area is a major factor driving concentrations in your study area, so it seems a large omission to not include it in your analysis. Is it simply that the agricultural area was not included in the database that you utilized? If so, could you obtain that information through other sources of land-use data? It is possible that including agricultural area in your analysis would significantly change the findings of your analysis regarding significant drivers of surface water concentrations.*

Agreed and changed accordingly, see response 1 above.

*p. 6, ll. 14-18 In your methods, you mention that for each well, you average concentrations for each monitoring well (at individual monitoring screens) for all sampling dates. You also mention that the groundwater data is from the period 1910-2013 more than 100 years. I would assume that there could have been significant changes in groundwater quality over that period, and that the temporal patterns of change could have differed across the study period. Accordingly, is it correct to combine all sampling data across this 100-year period, or in doing so are you conflating spatial and temporal differences across the study area?*

Agreed and changed accordingly, see response 2 above.

*p. 8, ll. 20-30 You do not discuss here how you dealt with issues of collinearity among the explanatory variables. For example, there are clearly high correlations (r>0.60) among some of the groundwater solute concentrations (particularly with regard to $HCO_3$). With this being the case, how do you (from a quantitative perspective) make decisions regarding inclusion in the multiple linear regression model? For example, in your MLR equation for TP, you include both $HCO_3(GW)$ and TP(GW), although your correlation table in Table 1 shows a reasonably high collinearity (r=0.68) between these two variables. How do you justify use of both of them in the MLR equation?*

Agreed and changed accordingly, see response 3 above.

p. 12, ll. 22-23 You say here that ammoniums correlates more strongly with TN than nitrate and conclude that ammonium is therefore likely the main form of TN in the

study area. When I look at Fig. 5, however, it appears that nitrate is likely the dominant form of N in the ice-pushed ridge area (5) and possibly the Vecht Lakes area (4). It might be more useful to discuss the actual variations among locations (and reasons why), rather than just to cite the simple regression results.

We agree that information should be added about the spatial differences in TN partitioning. We've changed the text of line 22-23 into:

"Surface water TN correlated more closely to $NH_4$ (0.77) than to $NO_3$ (0.57). This reflects that $NH_4$ is the dominant form of TN in the study area as a whole. This is especially true for the upconing area and the Central Holland area (see also Figure 9). The $NO_3$ and $NH_4$ contributions to TN are about equal in the Vecht lakes area. For the Ice pushed ridge area, we expect a dominance of $NO_3$ in surface water (not shown in Figure 9 due to insufficient data) as was the case in the groundwater of that area, however there is only a limited amount of surface water that is draining the ice pushed ridge directly."

*p. 13, ll. 8-21 You discuss the results of the MLR analysis here, but do not reference the table that contains the results. Please include the table reference here.*

Agreed and changed accordingly.

*Fig. 5 It is very difficult to understand the variations in concentrations of solutes among locations in these figures due to the different concentration ranges from site to site. For example, for TN, all of the concentration ranges look very similar, simply because you scale the y-axis to include all of the outlier values for site #5. Is it important to include all of the outliers? I would recommend plotting these in such a way that you allow the reader to understand differences in median and interquartile range values, rather than prioritizing the representation of outliers.*

Agreed and changed accordingly as below.

[Figure]

*Table 1    For your correlation analysis, you should include the 1.0 values to show perfect correlation between two identical variables. This will help add structure to the table and make it easier to understand.*

Agreed and changed accordingly.